# BFL-SDWANTrust: Blockchain Federated-Learning-Enabled Trust Framework for Secure East–West Communication in Multi-Controller SD-WANs

**DOI:** 10.3390/s25165188

**Published:** 2025-08-21

**Authors:** Muddassar Mushtaq, Kashif Kifayat

**Affiliations:** 1Department of Computer Science, Air University, Islamabad 44000, Pakistan; 2College of Computing and Intelligent Systems, University of Khorfakkan, Sharjah 18119, United Arab Emirates; kashif.kifayat@ukf.ac.ae

**Keywords:** blockchain, deep learning, east–west communication, federated learning, interplanetary file system, multi-controller, software-defined networks

## Abstract

Software-Defined Wide-Area Networks (SD-WAN) efficiently manage and route traffic across multiple WAN connections, enhancing the reliability of modern enterprise networks. However, the performance of SD-WANs is largely affected due to malicious activities of unauthorized and faulty nodes. To solve these issues, many machine-learning-based malicious-node-detection techniques have been proposed. However, these techniques are vulnerable to various issues such as low classification accuracy and privacy leakage of network entities. Furthermore, most operations of traditional SD-WANs are dependent on a third-party or a centralized party, which leads to issues such single point of failure, large computational overheads, and performance bottlenecks. To solve the aforementioned issues, we propose a Blockchain Federated-Learning-Enabled Trust Framework for Secure East–West Communication in Multi-Controller SD-WANs (BFL-SDWANTrust). The proposed model ensures local model learning at the edge nodes while utilizing the capabilities of federated learning. In the proposed model, we ensure distributed training without requiring central data aggregation, which preserves the privacy of network entities while simultaneously improving generalization across heterogeneous SD-WAN environments. We also propose a blockchain-based network that validates all network communication and malicious node-detection transactions without the involvement of any third party. We evaluate the performance of our proposed BFL-SDWANTrust on the InSDN dataset and compare its performance with various benchmark malicious-node-detection models. The simulation results show that BFL-SDWANTrust outperforms all benchmark models across various metrics and achieves the highest accuracy (98.8%), precision (98.0%), recall (97.0%), and F1-score (97.7%). Furthermore, our proposed model has the shortest training and testing times of 12 s and 3.1 s, respectively.

## 1. Introduction

Software-Defined Wide-Area Networks (SD-WANs) have recently gained much attention due to their contributions in modern enterprise and cloud networking. SD-WANs ensure flexible and reliable programmable control over wide-area networks. They ensure easier configuration and management, which improves the performance of the network while simultaneously ensuring simple and efficient connectivity at multiple locations [1]. Communication between multiple SDN controllers over east–west interfaces plays a significant role in coordinated policy enforcement, routing, and control-plane synchronization across distributed domains [2]. However, these east–west interfaces are vulnerable to various issues such as data tampering, false policy injection, and unauthorized access, which affect the integrity of data and the privacy of network entities. The SDN-WANs are growing rapidly to meet the needs of various enterprises. Therefore, it is important to ensure trustworthy and reliable communication between the controllers to protect sensitive control-plane information [3]. However, these networks are vulnerable to various issues such as leakage of privacy of network entities, low security, and the presence of malicious entities [4], because of which the internal users are not comfortable sharing their data with other network entities. Similarly, external users are also not confident in joining and relying upon such networks. The malicious nodes disseminate faulty information in the network, which causes routing inconsistencies and affects the reliability of network services [5]. This raises concerns about the privacy of network entities and their sensitive and important information [6]. Therefore, a robust, privacy-preserving security framework is required that can efficiently detect malicious nodes, validate trust, and secure communication among SDN controllers in a decentralized manner.

During recent years, many studies have been conducted to solve the aforementioned issues of privacy leakage and the presence of malicious nodes in the SD-WANs while developing various ML/DL models [7,8,9]. Abdinasir et al. proposed a machine-learning-based malicious-node-detection mechanism, utilizes the capabilities of Logistic Regression (LR), Support Vector Machine (SVM), Random Forest (RF), K-nearest neighbour (KNN), and XGBoost [7]. Their model efficiently detects denial-of-service attacks and preserves the privacy of network entities. On similar lines, Manvitha et al. utilize the capabilities of deep learning algorithms for malicious-node detection while simultaneously ensuring effective and reliable quality of service-oriented routing [8]. This system uses a Deep Belief Network (DBN) that efficiently classifies malicious and benign entities within the network. The techniques mentioned above efficiently detect and remove malicious entities from SDNs; they are highly dependent on cloud servers and third-party trust management services such as controller authentication, policy synchronization, and behavioral analytics. The stated dependencies make these techniques vulnerable to single points of failure, low network scalability, and performance bottlenecks. To solve these issues, Satoshi Nakamoto proposed a decentralized Bitcoin-based mechanism in 2008 that authenticates all network entities and validates all their actions while utilizing the capabilities of decentralized consensus-based mechanisms [10]. Blockchain is a distributed network in which all network operations are distributed among network entities, without relying on any centralized or third-party [11,12,13], as shown in Figure 1. The blockchain utilizes the capabilities of various consensus algorithms and hashing techniques to establish trust among globally distributed network entities and ensure the integrity of their transferred data [14,15,16,17]. Besides this, it uses different cryptographic algorithms that convert the input data into fixed-sized ciphertexts, which helps in preserving the privacy of entities and ensures tamper resistance [18,19]. Various digital smart contracts are also integrated in a blockchain network that contains predefined business rules and conditions [20]. These smart digital contracts help in ensuring transparent and efficient execution of agreements without any involvement of a centralized party [21].

The blockchain-based security frameworks for SDNs proposed so far utilize various machine learning and deep learning techniques to classify malicious and legitimate nodes based on their inter-controller communication behaviors [22,23]. These techniques rely on various algorithms such as LR, SVM, RF, KNN, and XGBoost. All these techniques have a centralized model training structure in which east–west traffic logs, controller behavior analytics, and policy synchronization metadata from multiple domains are aggregated at a central server for training and inference. In these techniques, controllers need to transmit the sensitive operational and telemetry data of the network to a centralized entity, which makes them susceptible to issues such as privacy leakage and low detection accuracy. To solve these issues, we propose a Blockchain Federated-Learning-Enabled Trust Framework for Secure East–West Communication in Multi-Controller SD-WANs (BFL-SDWANTrust) in multi-controller SD-WANs [24,25,26,27]. In our proposed BFL-SDWANTrust, the edge SDN controllers are responsible for local training of malicious-node detection and trust evaluation models using their east–west interface traffic and policy synchronization metadata. After local model training, only essential model parameters such as encrypted gradients and weights are shared with a central aggregator unit for the generation of the federated global model [28,29]. In this way, the sensitive control-plane data and east–west communication logs are not shared outside the network, which helps in ensuring data privacy [30]. After the aggregation of the global trust model, it is sent to all participating SDN controllers, which ensures accurate and generalized trust evaluation across all east–west interfaces. In this way, our proposed model efficiently enhances the robustness and adaptability of trust evaluation in multi-controller SD-WAN environments [31]. This collaborative learning mechanism also enhances detection accuracy by utilizing diverse network-wide data without compromising the privacy of network entities. Besides this, our proposed model also ensures real-time malicious-node detection and significantly reduces response latency, as edge SDN controllers train their local trust models closer to the source of east–west traffic and control-plane interactions [32]. Furthermore, the transport layer security (TLS) channels between SDN controllers protect the SDN network against various attacks [33]. Although TLS provides mutual authentication for secure communication between SDN controllers, it cannot by itself prevent certain dynamic impersonation threats in distributed environments. For instance, if a legitimate controller is compromised after the TLS handshake or if its credentials are stolen, a rogue node can maintain a valid TLS session while exhibiting deviant behaviors during synchronization or policy enforcement. Our model targets such behaviors by analyzing control message patterns, response consistency, and policy compliance, factors outside the TLS scope. The impersonation threats described in our work refer to post-authentication trust degradation, wherein a once-authenticated controller begins to disseminate conflicting or faulty updates. This threat model reflects real-world zero-trust architectures where behavior-based trust scoring complements cryptographic guarantees, ensuring defense even when authentication channels are subverted post-establishment. In this way, SDN networks are still vulnerable to multiple attacks such as injection of falsified policy updates, misconfiguration propagation, replayed synchronization messages, and rogue controller impersonation. Our model addresses these risks by evaluating the trustworthiness of each controller through reputation scoring, behavioral validation, and real-time synchronization monitoring. The framework also identifies malicious controllers masquerading as legitimate nodes during collaborative processes, a threat not mitigated by transport layer security alone. The detection of such behaviors is central to our federated trust evaluation model. The major contributions of the proposed BFL-SDWANTrust are given as follows:A BFL-SDWANTrust model is proposed that ensures reliable and privacy-preserving distributed local model training across multiple SDN controllers while utilizing the capabilities of Convolution Neural Networks (CNN) and SVM classifiers.A distributed blockchain network is used to remove dependence on centralized authorities and efficiently solve various issues such as single points of failure, performance bottlenecks, and operational overheads in controller-to-controller communication.CNN and SVM are integrated to extract spatial-temporal patterns and perform robust classification of east–west traffic behaviors. This hybrid technique not only enhances anomaly detection accuracy but also the controller trust prediction in heterogeneous and dynamic SD-WAN environments.

Table 1 provides the list of abbreviations used in this paper. The remainder of the manuscript is organized as follows: Section 2 presents a detailed review of related work in SD-WAN trust management and blockchain-based security. Section 3 introduces the BFL-SDWANTrust architecture, describing each of its four layers in detail. Section 4 and Section 5 outlines the experimental setup, discusses the evaluation metrics used to validate our model, and analyzes the experimental results. Section 6 concludes the paper and outlines potential directions for future work.

## 2. Related Work

Many studies have been conducted to ensure effective and reliable deployment of SDN for remote and dispersed locations. Alrashede et al. state that existing techniques are not able to secure the east–west interface and communication between SDN controller nodes, which compromises the security of the overall network [34]. Therefore, the authors propose a blockchain-based distributed mechanism that ensures distributed credential-based access control and encrypted security of data while utilizing the capabilities of the Ethereum network and customized blockchain protocols [35]. The encryption mechanism ensures that the data are encrypted with a particular key and can only be decrypted by the receiver node, which solves the issues of man-in-the-middle and false data injection attacks. Furthermore, the proposed model utilizes the SHA-256 algorithm for data integrity. The proposed model ensures effective and reliable node authentication with swift registration and a mutual authentication mechanism. This model is evaluated by comparing it with various state-of-the-art and benchmark schemes. The performance of all the models is evaluated while considering the factors of authentication time, decentralization, scalability, authentication, system latency, authentication latency, registration latency, throughput, and access control. However, the model of [34] is highly dependent on the Ethereum network, which leads to large network overhead, such as increased latency and operational costs. Furthermore, the system is not scalable as the scalability performance of the proposed model is not evaluated in terms of node churn and dynamic SDN environments, which affects the real-world applicability of this network. While the model in [34] relies on Ethereum for its blockchain operations, our proposed model uses a customized PoA consensus mechanism optimized for controller-to-controller coordination. Ethereum-based systems often suffer from gas costs, execution latency, and unnecessary mining overhead. Our lightweight PoA system removes these constraints by selecting validators from known, high-reputation controllers, thus reducing delay and energy consumption. Furthermore, our system integrates IPFS for data distribution and applies federated learning for real-time trust adaptation that is not used in [34]. These architectural decisions make our model more suitable for scalable, real-time SD-WAN deployments.

Mao et al. [36] proposed a blockchain and federated-learning-enabled security protocol that efficiently detects the attacks and protects the network against them. Furthermore, the capabilities of the federated learning mechanism were utilized, which solved the issue of data privacy. The federated learning mechanism ensured that no raw data of network nodes was shared outside the networks, and local model training was performed within the subnetwork. This not only provided a privacy-preserving network for SDN controller devices but also enhanced the accuracy of the attack detection process as more generalized models were used by all subnetworks for attack detection. The performance of this model was evaluated while considering the factors of data flow rate, accuracy, true positive rate, recall, true negative rate, and velocity updates. However, the model faced performance degradation as this model has a large computational overhead of federated learning, especially in resource-constrained SDN edge devices. Besides this, the model does not efficiently detect poisoned and malicious local models in the federated setup, which affects the integrity of global models. Ahmad et al. [37] proposed a decentralized attribute-based access control mechanism that removes centralized entities from the network and ensures effective and reliable cross-domain access control. This model utilizes the capabilities of group signature, which helps in the secure exchange of attribute information while simultaneously preserving the privacy of network entities within a cross-domain. Each domain has its own predefined rules and polices, which establish a decentralized network and reduce the dependence on global trust models (Table 2).

Furthermore, the authors use a distributed consensus algorithm that evaluates the practical feasibility of the proposed model while simultaneously ensuring secure and reliable exchange of healthcare information in the cross-domain network. The performance of this model is evaluated while considering the factors of several predicates, the number of attributes, model performance, and the execution time. However, this model is vulnerable to the issues of performance bottlenecks due to the large computational overhead of group signature schemes, especially when handling multiple attributes or predicates. Furthermore, the model has a limited dynamic policy adaptation mechanism, which limits the flexibility in rapidly evolving cross-domain networks. Abdelouahid et al. [38] proposed a blockchain-based decentralized architecture in which the network is divided into various domains. Each domain has its master controller that is responsible for interacting with the master controllers of other domains. All the interactions and transactions of a respective domain are validated by its master controller. After the validation of each block, the master controller adds the block to the blockchain network. Furthermore, the model utilizes the capabilities of the fading reputation mechanism that ensures effective and optimal detection of malicious nodes in the network. The model was evaluated while considering the factors of detection rate, execution time, number of switches, number of hosts, detection time, reputation, and number of observation intervals. However, all the operations of this model are highly dependent on the master controller of the respective domain, which causes the issue of a single point of failure within each domain, compromising the decentralization objective of the proposed model. Furthermore, the fading reputation mechanism is vulnerable to evasion tactics, as some malicious entities can act normally for a short time to avoid detection. Furthermore, Rahman et al. [39] proposed a machine learning and blockchain-enabled SDN that ensures network security while simultaneously preserving the privacy of entities and the flexibility of the network. The authors divide the whole network into various data planes and control planes, which helps in efficient and reliable management of the network with low computational overhead. Furthermore, their model utilizes the capabilities of a distributed blockchain network to ensure network security and privacy of network entities with the help of SHA-256 hashing and Keccak encryption techniques, respectively. The model uses long short-term memory and multilayer perceptron classifiers for estimation of network bandwidth, which helps in quick and intelligent decision-making while simultaneously satisfying high-bandwidth demand. The performance of the proposed model was evaluated via experimental evaluation in terms of network topology, bandwidth, throughput, node failure rate, computational delay, and bandwidth prediction. However, this model has a large computational overhead and network latency due to the training of LSTM and MLP classifiers, especially in real-time high-speed fifth-generation networks. Furthermore, the model does not have any mechanism to consider interoperability challenges between heterogeneous fifth-generation network components during deployment.

## 3. Proposed BFL-Model

Figure 2 shows the hierarchical and distributed architecture of SDN controllers, where the east–west communication represents the exchange of control information among SDN controllers deployed across a nationwide SD-WAN infrastructure. Building on this architecture, Figure 3 presents the proposed BFL-SDWANTrust framework, which is specifically designed to enhance the security and trustworthiness of east–west communication in such multi-controller SD-WAN environments. Our proposed model is divided into four layers: sensing layer, blockchain layer, fog layer, and cloud layer. Each layer plays an important role in establishing a decentralized trust framework by utilizing the capabilities of blockchain for immutable record management and consensus-based validation [40]. Furthermore, federated learning and cryptographic techniques are used for distributed malicious-node detection without sharing raw data and authentication of network entities, respectively. This layered architecture enhances the trust across east–west interfaces of multiple SDN controllers in the SD-WANs while simultaneously enhancing the accuracy of the detection mechanism.

### 3.1. Sensing Layer in East–West Interface Monitoring

In our proposed layered BFL-SDWANTrust network, the first layer is the sensing layer that is responsible for real-time environmental monitoring. Furthermore, this layer also extracts various trust-critical parameters from east–west traffic among distributed SD-WAN controllers $C_{i}$, where $C=\{C_{1},C_{2},\ldots ,C_{n}\}$ represents the set of SDN controllers that operate within a multi-domain SD-WAN.

In this layer, various system parameters such as packet delivery ratio (PDR), transaction latency, control message integrity, response consistency, and behavior anomaly scores are collected for evaluating network behavior [41]. These metrics are collected over time and multi-dimensional time-series datasets *M* are generated, as represented in Equation (Equation 1).(1)$$M=\{\left(t_{1},m_{1}\right),\left(t_{2},m_{2}\right),\ldots ,\left(t_{z},m_{z}\right)\}$$
where, $m_{z}$ is a vector of trust parameters observed at time $t_{z}$ and *z* represents the number of observed data intervals. The metrics are collected by each controller at a defined interval based on a configurable probing frequency $f_{p}$, which is formulated according to Equation (Equation 2).(2)$$f_{p}=\frac{1}{\Delta t}$$
where, Δt is the time gap between two successive metric samples. When all sensors or nodes perform normally, then there is a lower frequency, while the monitoring frequency is increased for further observation in case of any unusual activity in the network. Furthermore, we utilize the capabilities of asymmetric encryption and the Keccak-256 hashing algorithm for protecting controller data, as represented in Equation (Equation 3), which ensures secure communication between all layers [42].(3)$$\Psi_{enc}=H\left(Enc_{K_{pub}}\left(M\right)\right)$$
where, $Enc_{K_{pub}}\left(M\right)$ represents the encrypted metric dataset using the public key $K_{pub}$ of the corresponding blockchain verifier node and H(·) is a cryptographic Keccack-256 function, which ensures the integrity of data. After that, $\Psi_{enc}$ is sent to the blockchain verification layer for further trust evaluation. Furthermore, we utilize the capabilities of lightweight encoding techniques such as Context-Adaptive Binary Arithmetic Coding (CABAC) for compressing encrypted data. It helps in ensuring efficient transmission and reducing congestion across the east–west interfaces. We also use hybrid SD-WAN communication protocols such as gRPC and NETCONF/YANG for reliable controller-to-controller communication. These protocols not only ensure high throughput and low latency but also provide an efficient and reliable model-driven communication [43].

In our proposed model, all SD-WAN controllers are registered with a trust unit at the blockchain layer by providing their credentials, such as identity certificates, resource capabilities, policy definitions, and operational thresholds. When any controller sends trust-critical updates, it also adds its registration credentials, operational logs, and communication signatures. Upon receiving the data, the blockchain layer triggers the smart contracts to authenticate the sender controller by matching its credentials [44]. If both records match, then the sending controller is considered to be a legitimate node, and an acknowledgment message is sent back to the respective sensor node to ensure the delivery of the data packet. On the other hand, if both records do not match, then these controller devices are considered to be malicious and immediately removed from the network [45]. The whole process of the sensing layer is described in Algorithm 1.

Algorithm 1 is designed to monitor and secure communication between distributed SDN controllers $C=\{C_{1},C_{2},\ldots ,C_{n}\}$ within a multi-controller SD-WAN architecture. Each controller $C_{i}$ collects trust-critical parameters at defined intervals governed by a probing frequency $f_{p}$, which is the inverse of the time difference Δt between metric samples.
**Algorithm 1** Sensing layer algorithm for east–west interface trust evaluation**Require:** Set of SDN controllers $C=\{C_{1},C_{2},\ldots ,C_{n}\}$, Probing frequency $f_{p}=\frac{1}{\Delta t}$,   Threshold trust value θ, Public key $K_{pub}$ of blockchain node**Ensure:** Encrypted trust dataset $\Psi_{enc}$, Trust evaluation report $T_{r}$  1: Initialize metric set M=∅, timestamp t=0  2: **for** each controller $C_{i}\in C$ **do**  3:     Initialize local metric buffer $M_{i}=\emptyset$  4: **end for**  5: **while** network is active **do**  6:     **for** each controller $C_{i}\in C$ **do**  7:         Sample trust-critical parameters $m_{t}=\left[{\mathrm{PDR}}_{t},L_{t},{\mathrm{CMI}}_{t},RC_{t},BAS_{t}\right]$  8:         Record timestamp t←t+Δt  9:         Append $\left(t,m_{t}\right)$ to $M_{i}$  10:        **if** anomaly detected in $m_{t}$ **then**  11:            Increase probing frequency $f_{p}\leftarrow f_{p}+\delta$  12:        **else**  13:            Restore probing frequency $f_{p}\leftarrow \frac{1}{\Delta t}$  14:         **end if**  15:         Compress $m_{t}$ using CABAC: ${\hat{m}}_{t}\leftarrow \mathrm{CABAC}\left(m_{t}\right)$  16:         Encrypt ${\hat{m}}_{t}$: $E_{t}\leftarrow Enc_{K_{pub}}\left({\hat{m}}_{t}\right)$  17:         Generate integrity hash: $h_{t}\leftarrow H\left(E_{t}\right)$  18:         Create secure packet: $\Psi_{enc}^{t}\leftarrow \left\{E_{t},h_{t}\right\}$  19:         Send $\Psi_{enc}^{t}$ to blockchain verifier  20:     **end for**  21:     **for** each received $\Psi_{enc}^{t}$ at blockchain layer **do**  22:         Decrypt $E_{t}$ using $K_{priv}$, validate hash $h_{t}$  23:         **if** $H\left(E_{t}\right)=h_{t}$ **then**  24:            Validate sender identity via smart contract *SC*  25:            **if** identity ∈ registry and signature matches **then**  26:                Approve trust record: $T_{r}\leftarrow T_{r}\cup \left\{C_{i},t,m_{t}\right\}$  27:                Send ACK to $C_{i}$  28:            **else**  29:               Mark $C_{i}$ as malicious  30:                Remove $C_{i}$ from network set *C*  31:            **end if**  32:         **else**  33:            Reject data: Integrity violation  34:         **end if**  35:     **end for**  36: **end while**  37: **return** Final encrypted trust dataset $\Psi_{enc}$ and report $T_{r}$

These parameters include the packet delivery ratio (PDR), transaction latency $L_{t}$, control message integrity (CMI), response consistency (RC) and behavior anomaly score (BAS), collectively represented as the trust metric vector $m_{t}$ at time *t*. Each controller maintains a local metric buffer $M_{i}$ which is populated with timestamped observations $\left(t,m_{t}\right)$. If any anomaly is detected in the observed metrics, the algorithm adaptively increases the probing frequency $f_{p}$ by a constant value δ to enhance monitoring sensitivity; otherwise, it restores the default sampling frequency. To reduce transmission overhead, each metric vector is compressed using Context-Adaptive Binary Arithmetic Coding (CABAC), producing a compressed vector ${\hat{m}}_{t}$.

After that, the encrypted data are generated using asymmetric encryption $Enc_{K_{pub}}\left(\cdot \right)$, where $K_{pub}$ is the public key of the blockchain verifier node. This encrypted metric data $E_{t}$ is then hashed using a cryptographic Keccak-256 hash function H(·) to generate a secure digest $h_{t}$, ensuring data integrity. The final encrypted and hashed pair is encapsulated into a secure packet $\Psi_{enc}^{t}=\left\{E_{t},h_{t}\right\}$, which is transmitted to the blockchain layer for validation. Upon receipt, the blockchain layer decrypts the packet using the corresponding private key $K_{priv}$ and validates the hash to confirm data integrity. If the hash matches, a smart contract *SC* is triggered to verify the sender controller’s credentials. If the identity and signature of the controller match those in the blockchain registry, the trust record is appended to the global trust report $T_{r}$ and an acknowledgment is sent back to the controller. Conversely, if the identity verification fails, the controller is labeled as malicious and is excluded from the trusted network set *C*. If the hash check fails, indicating a tampered message, the packet is rejected. This algorithm ensures that encrypted and trustworthy behavioral data are continuously monitored and validated, preserving the confidentiality, integrity, and availability of east–west communications in a decentralized SD-WAN environment.

### 3.2. Blockchain Layer for Secure East–West Interface Communication

The second layer of our proposed BFL-SDWANTrust framework is the blockchain layer, which plays an important role in establishing trust and data immutability. Furthermore, this layer utilizes the capabilities of a blockchain decentralized ledger to ensure effective and secure east–west interface communications among SDN controllers in a multi-controller SD-WAN environment. This layer comprises various SDN controllers $C_{i}$, where each controller is responsible for managing its regional domain and communicates with other controllers over the east–west interface [46]. In our proposed model, the blockchain network is deployed over the controller network to maintain a tamper-resistant ledger that captures all communication data and routing coordination updates. In this layer, the blockchain smart contracts are also deployed, which ensure automated trust evaluation. Furthermore, we use the Proof-of-Authority (PoA) consensus algorithm to establish consensus among distributed controller nodes. Each SDN controller $C_{i}$ is registered with the blockchain ledger by providing its credentials and cryptographic identity to the trusted node [2]. All the real-time transactions among controller nodes are validated by the PoA consensus algorithm. The block-wise control data $R_{C}^{t}$ collected at time *t* is represented as Equation (Equation 4).(4)$$R_{C}^{t}=\left\{r_{1}^{t},r_{2}^{t},\ldots ,r_{p}^{t}\right\}$$
where, $r_{k}^{t}$ is the trust communication packet from controller *k* at time *t* and *p* is the number of communication packets aggregated for ledger update. Furthermore, the capabilities of digital signatures and Keccak-256 techniques are used to ensure the security and non-repudiation of the control data. In our proposed model, a secure hash $H_{blk}$ is generated over the block content using the private key $SK_{i}$ of controller $C_{i}$, as shown in the Equation (Equation 5).(5)$$H_{blk}=\mathrm{Hash}\left(R_{C}^{t}∥SK_{i}\right)$$
where, $H_{blk}$ is the unique identifier of the control data block, $R_{C}^{t}$ is the block of east–west control packets and $SK_{i}$ is the private cryptographic key of controller $C_{i}$, which ensures that only authorized controllers can issue valid transactions. Furthermore, the PoA consensus algorithm ensures that only trusted and pre-authorized SDN controllers can validate east–west communication transactions and add the blocks to the distributed blockchain ledger [47]. Each interested SDN controller $C_{i}$ expresses its interest in becoming a validator for the current transactions. However, only controllers with a high reputation $\rho_{i}$ and consistent historical behavior $\tau_{i}$ are selected for transaction validation and adding the block. After that, a weighted scoring function is used to rank the interested nodes, and only the controllers with high values are selected as validators. While the architecture shown in Figure 2 indicates logical centralization, the participating SDN controllers operate in federated domains with autonomy in runtime policy adaptation based on local conditions such as latency, route availability, and domain-specific service level agreements (SLAs). The logical controller acts as a high-level entity but does not override the local operational policies and behavioral thresholds enforced by each domain controller. Therefore, runtime metrics like policy enforcement accuracy and responsiveness can vary across controllers due to environmental and network conditions, justifying the differentiation in reputation scores. Our reputation framework is designed to capture such deviations and ensure controller behaviors are continuously validated in dynamic east–west interactions, even under logical central governance. The selection probability $P_{auth}\left(C_{i}\right)$ for a controller $C_{i}$ is calculated using Equation (Equation 6).(6)$$P_{auth}\left(C_{i}\right)=\frac{\omega_{1}\cdot \rho_{i}+\omega_{2}\cdot \tau_{i}}{\sum_{j=1}^{q}\left(\omega_{1}\cdot \rho_{j}+\omega_{2}\cdot \tau_{j}\right)}$$
where, $\rho_{i}$ represents the trust reputation of controller $C_{i}$, which is computed based on its audit compliance, policy enforcement accuracy, and network uptime. The variable $\tau_{i}$ shows the normalized number of successful transaction validations completed by controller $C_{i}$ over a recent time. The weights $\omega_{1}$ and $\omega_{2}$ are important factors that contribute to trust, reputation, and transaction history, respectively. The denominator computes the total weighted score of all *q* interested controllers. In our proposed model, the reputation $\rho_{i}$ of an SDN controller shows its historical compliance with network policies, responsiveness in synchronization, and availability. While policies are initially defined by administrators, their enforcement and consistency across east–west interfaces vary due to latency, partial failures, or configuration drift. Our trust framework evaluates these runtime behaviors to compute $\rho_{i}$, ensuring that validators in the blockchain layer are selected based not only on configuration but also on operational fidelity. This differentiation is crucial in dynamic SD-WAN environments where misbehaving or compromised controllers can propagate faulty policies. Once the validator nodes are selected based on Equation (Equation 6), the transaction verification process begins. Each selected validator $C_{k}$ verifies the transaction set *T* and casts a vote on its own. A consensus is reached when the number of approval votes $V_{yes}$ exceeds from predefined threshold $\theta_{cons}$, as shown in Equation (Equation 7).(7)$$\sum_{k=1}^{v}v_{k}=V_{yes}\geq \theta_{cons}$$
where, $v_{k}\in \left\{0,1\right\}$ is the binary vote of validator $C_{k}$, where 1 and 0 represent approval and rejection, respectively. The variable *v* represents the total number of validators in the current round, and $\theta_{cons}$ is the predefined minimum number of approvals that are required for adding a block to the blockchain ledger. If the consensus threshold is met, the verified block $B_{verified}$ is added to the blockchain [48]. The process of adding a block is protected with asymmetric encryption and Keccak-256 hashing techniques [49]. This hash is made with all the approved transactions *T* and the signatures of the validators $\Sigma_{v}$, as shown in Equation (Equation 8).(8)$$H_{commit}=H\left(T\;\|\;\Sigma_{v}\right)$$
where, H(·) is a secure cryptographic hash function, *T* is the concatenated list of verified transactions and $\Sigma_{v}$ represents the combined digital signatures of all approving validators. This hash ensures the integrity of the entire block, which ensures that the transaction set and validators cannot be tampered with or altered. In this way, the PoA consensus algorithm ensures that only trustworthy and high-performing controllers participate in consensus, which ensures the reliability, scalability, and decentralization of the proposed BFL-SDWANTrust blockchain layer [50].

Furthermore, we utilize the capabilities of an external data storage platform, InterPlanetary File System (IPFS), to solve the issue of the high cost of blockchain storage. In our proposed model, only the hash of the east–west control data is stored on the blockchain, and the original data is stored on distributed nodes of IPFS. Our proposed model utilizes the capabilities of IPFS to offload storage from the blockchain, while maintaining verifiability of east–west communication logs. Each control data block, containing synchronization updates and policy actions, is hashed using Keccak-256 (with less transaction latency than SHA-256) and stored on IPFS. The corresponding hash is written to the blockchain, ensuring immutability. Access to these records requires signature verification of the requesting entity using smart contracts. This layered mechanism ensures that only registered and verified controllers can read or append to the communication log, preventing unauthorized access or tampering, thereby securing the east–west channel beyond standard encryption mechanisms. First, the controller calculates the hash of data and sends it to the IPFS storage platform, and the hash is stored on the blockchain ledger, as shown in Equation (Equation 9).(9)$$H_{IPFS}=\mathrm{Hash}\left(R_{C}^{t}\right)\to \mathrm{IPFS}$$
where, $H_{IPFS}$ denotes the hash of data that is used by IPFS to identify and retrieve the control data. After that, IPFS divides the data into fixed-size packets of 256 kB and stores these packets at its globally distributed nodes. The processes of segmentation and distribution of control data are expressed in Equation (Equation 10).(10)$$D_{C_{i}}=\overset{w}{\underset{k=1}{⋃}}\Pi_{k},\;\Pi_{k}\in N_{IPFS}$$
where, $D_{C_{i}}$ is the data from controller $C_{i}$, $\Pi_{k}$ is a segmented packet of 256 kB and $N_{IPFS}$ denotes the global IPFS node network. When a controller requests data from IPFS, it must provide the hash of the data stored on the blockchain. The IPFS verifies the authenticity of data with the hash and collects the segmented data from its distributed nodes. After that, all the segmented data are aggregated by IPFS and sent to the controller. Upon receiving the data, the controller again calculates the hash of the data and compares it with the stored blockchain hash. It helps in verifying the integrity of data, as represented in Equation (Equation 11).(11)$$H_{verify}=\mathrm{Hash}\left(\sum_{k=1}^{w}\Pi_{k}\right)$$

If $H_{verify}=H_{IPFS}$, an acknowledgment is sent to ensure the integrity of data. Otherwise, the smart contract triggers an arbitration process to resolve the dispute. In this way, the blockchain layer of our proposed model ensures that only authenticated controllers and federated nodes can access the data. All entities must register through smart contracts to be considered legitimate participants. In the proposed model, the capabilities of digital signature validation are utilized to ensure the access control mechanism. When a controller $C_{i}$ or a federated learner node requests access to data, its digital signature is verified. The access decision $A_{ctrl}$ is defined in Equation (Equation 12).(12)$$A_{ctrl}=\left\{\begin{array}{cc} 1, & \mathrm{if}\;\mathrm{Sig}(PK_{node})\;\mathrm{is}\;\mathrm{valid}\\ 0, & \mathrm{otherwise} \end{array}\right.$$
where, $PK_{node}$ is the public key of the requesting entity and $\mathrm{Sig}(PK_{node})$ is its submitted digital signature. If the signature is valid, only then is access given; otherwise, it is denied. The whole process of secure block formation, IPFS integration, consensus validation, and access control is described in Algorithm 2.
**Algorithm 2** Blockchain layer algorithm for secure east–west communication**Require:** East–west control packets $R_{C}^{t}$, controller set $\left\{C_{1},C_{2},\ldots ,C_{n}\right\}$**Ensure:** Verified block $B_{verified}$, access control decision $A_{ctrl}$  1: **for all** controller $C_{i}\in \left\{C_{1},C_{2},\ldots ,C_{n}\right\}$ **do**  2:     Collect trust communication packets $r_{k}^{t}$  3:     Form control data block $R_{C}^{t}=\left\{r_{1}^{t},r_{2}^{t},\ldots ,r_{p}^{t}\right\}$  4:     Compute hash $H_{blk}=\mathrm{Hash}\left(R_{C}^{t}\;\|\;SK_{i}\right)$  5:     Store $H_{blk}$ for block verification  6: **end for**  7: **PoA Validator Selection**  8: **for all** interested controller $C_{i}$ **do**  9:     Compute reputation score $\rho_{i}$ and transaction history $\tau_{i}$  10:   Compute selection probability:          $P_{auth}\left(C_{i}\right)=\frac{\omega_{1}\cdot \rho_{i}+\omega_{2}\cdot \tau_{i}}{\sum_{j=1}^{q}\left(\omega_{1}\cdot \rho_{j}+\omega_{2}\cdot \tau_{j}\right)}$  11: **end for**  12: Select top-*v* validators based on $P_{auth}\left(C_{i}\right)$  13: **Consensus Voting**  14: **for all** validator $C_{k}$ **do**  15:    Verify transaction set *T*  16:    Cast vote $v_{k}\in \left\{0,1\right\}$  17: **end for**  18: Compute total approvals: $V_{yes}=\sum_{k=1}^{v}v_{k}$  19: **if**
$V_{yes}\geq \theta_{cons}$
**then**  20:    Compute commit hash $H_{commit}=H\left(T\;\|\;\Sigma_{v}\right)$  21:    Form $B_{verified}$ and append to blockchain  22: **else**  23:    Discard block and log failure  24: **end if**  25: **IPFS Storage Integration**  26: Compute IPFS hash: $H_{IPFS}=\mathrm{Hash}\left(R_{C}^{t}\right)\to \mathrm{IPFS}$  27: Split $R_{C}^{t}$ into chunks $\Pi_{k}\in N_{IPFS}$  28: $D_{C_{i}}={⋃}_{k=1}^{w}\Pi_{k}$  29: Store $\Pi_{k}$ on distributed IPFS nodes  30: **Data Retrieval and Integrity Check**  31: Controller sends retrieval request with $H_{IPFS}$  32: IPFS collects $\Pi_{k}$, aggregates data and returns  33: Controller computes verification hash: $H_{verify}=\mathrm{Hash}\left(\sum_{k=1}^{w}\Pi_{k}\right)$  34: **if** 
$H_{verify}=H_{IPFS}$ 
**then**  35:     Send acknowledgment and continue  36: **else**  37:     Trigger arbitration process via smart contract  38: **end if**  39: **Access Control via Signature Validation**  40: **for all** access requests by $C_{i}$ or federated nodes **do**  41:     Verify $\mathrm{Sig}(PK_{node})$  42:     **if** $\mathrm{Sig}(PK_{node})$ is valid **then**  43:         Set $A_{ctrl}\leftarrow 1$  44:     **else**  45:         Set $A_{ctrl}\leftarrow 0$  46:     **end if**  47: **end for**

Algorithm 2 is designed to ensure secure, decentralized, and tamper-resistant communication among SDN controllers operating in a multi-controller SD-WAN environment. The algorithm begins by iterating over each controller $C_{i}\in \left\{C_{1},C_{2},\ldots ,C_{n}\right\}$, where *n* is the total number of SDN controllers participating in the east–west trust evaluation. Each controller collects trust communication packets denoted as $r_{k}^{t}$, where *k* represents the index of the packet and *t* the current time instance. These packets are aggregated to form the control data block $R_{C}^{t}=\left\{r_{1}^{t},r_{2}^{t},\ldots ,r_{p}^{t}\right\}$, with *p* being the total number of collected packets. A cryptographic hash of the block is then generated using the controller’s private signing key $SK_{i}$, represented as $H_{blk}=\mathrm{Hash}\left(R_{C}^{t}\;\|\;SK_{i}\right)$, where ‖ denotes concatenation. This hash serves as the fingerprint of the block and is preserved for verification during consensus. Subsequently, the algorithm proceeds with validator selection under the Proof-of-Authority (PoA) consensus mechanism, where interested controllers propose themselves as validators. Each candidate controller computes a reputation score $\rho_{i}$, based on historical behavior and transaction history $\tau_{i}$. These values are weighted using coefficients $\omega_{1}$ and $\omega_{2}$, respectively, to calculate the validator selection probability $P_{auth}\left(C_{i}\right)=\frac{\omega_{1}\cdot \rho_{i}+\omega_{2}\cdot \tau_{i}}{\sum_{j=1}^{q}\left(\omega_{1}\cdot \rho_{j}+\omega_{2}\cdot \tau_{j}\right)}$, where *q* denotes the number of interested validators. The top *v* controllers with the highest $P_{auth}$ values are selected as validators.

Once validators are selected, each validator $C_{k}$ verifies the transaction set *T* and casts a binary vote $v_{k}\in \left\{0,1\right\}$, where 1 indicates acceptance. The total number of affirmative votes is computed as $V_{yes}=\sum_{k=1}^{v}v_{k}$. If the approval count $V_{yes}$ meets or exceeds the predefined consensus threshold $\theta_{cons}$, a commit hash is generated as $H_{commit}=H\left(T\;\|\;\Sigma_{v}\right)$, where $\Sigma_{v}$ represents the aggregated validator votes. The verified block $B_{verified}$ is then appended to the blockchain ledger. To optimize storage and enhance decentralization, the algorithm incorporates the IPFS. It computes an IPFS hash $H_{IPFS}=\mathrm{Hash}\left(R_{C}^{t}\right)\to \mathrm{IPFS}$, splits the data block into chunks $\Pi_{k}\in N_{IPFS}$ and distributes them across IPFS nodes such that $D_{C_{i}}={⋃}_{k=1}^{w}\Pi_{k}$, where *w* is the number of chunks. Retrieval is performed using $H_{IPFS}$ and the aggregated data are verified by recomputing the hash $H_{verify}=\mathrm{Hash}\left(\sum_{k=1}^{w}\Pi_{k}\right)$. If $H_{verify}=H_{IPFS}$, the data are deemed valid; otherwise, a smart contract is triggered for arbitration. Finally, access control is enforced by verifying the digital signature $\mathrm{Sig}(PK_{node})$ of any requesting node. If valid, access control status is set to $A_{ctrl}\leftarrow 1$; otherwise, it is denied by setting $A_{ctrl}\leftarrow 0$. This process ensures both trustworthiness and data integrity in east–west controller communications.

### 3.3. Fog Layer in Multi-Controller SD-WAN

The fog layer of our proposed model plays an important role in the secure and distributed learning architecture of the BFL-SDWANTrust framework. At the fog layer, each VM is associated with its respective east–west controller node and is responsible for processing traffic metadata, flow records, and inter-controller communications [51]. Each virtual machine trains a local malicious-node-detection model while utilizing the capabilities of CNN and SVM classifiers with a federated learning mechanism. The input dataset $D_{ctrl}^{t}$ at time *t* for a controller $C_{i}$, gathered from east–west links, is defined according to Equation (Equation 13).(13)$$D_{ctrl}^{t}=\left\{d_{1}^{t},d_{2}^{t},\ldots ,d_{n}^{t}\right\}$$
where, $d_{k}^{t}$ represents the traffic flow feature vector of *k* that is observed on east–west traffic at time *t* and *n* is the total number of observed interactions during that timespan. These data are not shared outside the local fog network and are kept within the virtual machine server, which helps in preserving the privacy of network entities. Each VM trains a local malicious-node-detection model $T_{i}$ using lightweight neural network architectures, such as CNN and SVN, which helps in reliable and accurate malicious-node detection. The local model update for controller node $C_{i}$ is performed using a gradient descent step, as shown in Equation (Equation 14).(14)$$T_{i}^{t+1}=T_{i}^{t}-\mu \cdot \nabla L\left(T_{i}^{t},D_{i}\right)$$
where, $T_{i}^{t}$ represents the current local model of controller $C_{i}$ at iteration *t*, μ is the learning rate, ∇L(·) is the gradient of the loss function and $D_{i}$ is the dataset collected by controller $C_{i}$ from east–west traffic telemetry. On the other hand, the training of CNN-based models involves convolutional and fully connected layers that are optimized while utilizing the capabilities of backpropagation [52]. The model updates are updated according to Equation (Equation 15).(15)$$T_{\mathrm{CNN},i}^{t+1}=T_{\mathrm{CNN},i}^{t}-\mu \cdot \frac{1}{B}\sum_{j=1}^{B}\nabla L_{\mathrm{CNN}}\left(f_{\theta }\left(x_{j}\right),y_{j}\right)$$
where, $f_{\theta }$ is the CNN function parameterized by weights θ, $x_{j}$ and $y_{j}$ are the *j*-th input sample and its label, *B* is the batch size and $L_{\mathrm{CNN}}$ is the cross-entropy loss used for classification. The main purpose of SVM-based models is to minimize the hinge loss function. The update rule is expressed in Equation (Equation 16).(16)$$T_{\mathrm{SVM},i}^{t+1}=T_{\mathrm{SVM},i}^{t}-\mu \cdot \left(\lambda \cdot T_{\mathrm{SVM},i}^{t}-y_{j}x_{j}\cdot I\left(y_{j}\cdot \langle T_{\mathrm{SVM},i}^{t},x_{j}\rangle <1\right)\right)$$
where, λ is the regularization parameter, 〈·,·〉 represents the dot product between the weight vector and the input, I(·) is the indicator function, which has 1 output if the condition is true and $y_{j}\in \left\{-1,1\right\}$ is the class label of the sample $x_{j}$. These local updates ensure that each controller node can train an effective and privacy-preserving malicious-node-detection model while utilizing the data patterns of its respective east–west communication [53]. Furthermore, our proposed model uses asymmetric encryption to ensure that only authenticated and high-trust controllers contribute updates to the global trust model. Each VM digitally signs its locally trained model using the private key $\kappa_{priv}^{i}$ of controller $C_{i}$, as represented in Equation (Equation 17).(17)$$\Sigma_{i}=\mathrm{Sign}\left(T_{i}^{t+1},\kappa_{priv}^{i}\right)$$
where, $\Sigma_{i}$ is the digital signature of the source for the model. After transmission of the model to the blockchain layer for global aggregation and verification, each update is checked using the public key $\kappa_{pub}^{i}$ of respective $C_{i}$ and validated according to Equation (Equation 18).(18)$$\mathrm{Validate}\left(\Sigma_{i},\kappa_{pub}^{i}\right)=\left\{\begin{array}{cc} \mathrm{Accepted}, & \mathrm{if}\;T_{i}^{t+1}\;\mathrm{is}\;\mathrm{verified}\\ \mathrm{Rejected}, & \mathrm{otherwise} \end{array}\right.$$

This trust-preserving mechanism ensures that only legitimate east–west controller VMs can contribute to the aggregated model. In this way, our proposed BFL-SDWANTrust framework enhances the robustness and security of model convergence across the SD-WANs by ensuring only authenticated controllers have access to model updates [54]. The entire process of local model training at the fog layer is described in Algorithm 3. Algorithm 3 extends beyond conventional federated learning by incorporating domain-specific features for east–west trust evaluation. It integrates hybrid CNN+SVM training on localized east–west telemetry data, followed by digital signature-based authentication to ensure that only verified controllers contribute to the global model. Additionally, the model includes anomaly-aware sampling, encryption, and validation layers, tailored for decentralized controller networks. These additions distinguish our architecture from generic federated learning pipelines and address security-specific needs of SD-WAN controller planes.
**Algorithm 3** Local model training at fog layer**Require:**  1: Set of controllers $C=\{C_{1},C_{2},\ldots ,C_{n}\}$  2: Time step *t*, Learning rate μ, Batch size *B*, Regularization parameter λ  3: Dataset $D_{ctrl}^{t}=\left\{d_{1}^{t},d_{2}^{t},\ldots ,d_{n}^{t}\right\}$ from east–west traffic telemetry  4: Initial local malicious detection models:  5: CNN-based model $T_{\mathrm{CNN},i}^{t}$, SVM-based model $T_{\mathrm{SVM},i}^{t}$, General local model $T_{i}^{t}$  6: Controller private keys $\kappa_{priv}^{i}$ and public keys $\kappa_{pub}^{i}$**Ensure:**  7: Updated and signed local models $T_{i}^{t+1}$ with digital signatures $\Sigma_{i}$  8: Prepared model updates for secure transmission to the blockchain layer  9: **for** each controller node $C_{i}\in C$ **do**  10:     **Step 1: Data Collection**  11:     Load local traffic data $D_{ctrl}^{t}$ from east–west interface flows  12:     Extract feature vectors $d_{k}^{t}\in D_{ctrl}^{t}$  13:     **Step 2: Local Model Initialization**  14:     Retrieve previous local models $T_{i}^{t},T_{\mathrm{CNN},i}^{t},T_{\mathrm{SVM},i}^{t}$  15:     **Step 3: Gradient Computation for General Model**  16:     Compute gradient of loss: $\nabla L(T_{i}^{t},D_{ctrl}^{t})$  17:     **Step 4: General Model Update**  18:     Update local model weights:  19:                $T_{i}^{t+1}\leftarrow T_{i}^{t}-\mu \cdot \nabla L\left(T_{i}^{t},D_{ctrl}^{t}\right)$  20:     **Step 5: CNN Model Training**  21:     **For** each batch j=1…B **do**  22:                Extract batch samples $\left(x_{j},y_{j}\right)$  23:                Compute CNN gradient:  24:                     $\nabla L_{\mathrm{CNN}}\left(f_{\theta }\left(x_{j}\right),y_{j}\right)$  25:     **end for**  26:     Update CNN model:  27:                $T_{\mathrm{CNN},i}^{t+1}\leftarrow T_{\mathrm{CNN},i}^{t}-\mu \cdot \frac{1}{B}\sum_{j=1}^{B}\nabla L_{\mathrm{CNN}}\left(f_{\theta }\left(x_{j}\right),y_{j}\right)$  28:     **Step 6: SVM Model Training with Hinge Loss**  29:     **For** each sample $\left(x_{j},y_{j}\right)\in D_{ctrl}^{t}$ **do**  30:                Compute indicator function:  31:                     $I(y_{j}\cdot \langle T_{\mathrm{SVM},i}^{t},x_{j}\rangle <1)$  32:                Calculate SVM gradient component:  33:                     $g_{j}=\lambda \cdot T_{\mathrm{SVM},i}^{t}-y_{j}x_{j}\cdot I\left(\cdots \right)$  34:     **end for**  35:     Aggregate gradient over dataset and update:  36:                $T_{\mathrm{SVM},i}^{t+1}\leftarrow T_{\mathrm{SVM},i}^{t}-\mu \cdot \sum_{j}g_{j}$  37:     **Step 7: Digital Signature Creation**  38:     Sign updated model using private key:  39:                $\Sigma_{i}\leftarrow \mathrm{Sign}\left(T_{i}^{t+1},\kappa_{priv}^{i}\right)$  40:     **Step 8: Secure Transmission**  41:     Transmit $\left(T_{i}^{t+1},\Sigma_{i}\right)$ to blockchain layer  42:     **Step 9: Signature Verification**  43:     Verify signature with public key:  44:                $\mathrm{Validate}(\Sigma_{i},\kappa_{pub}^{i})$  45:     **if** validation accepted **then**  46:         Accept model update for global aggregation  47:     **else**  48:         Reject update, log security alert  49:     **end if**  50: **end for**

Algorithm 3 is designed to perform local training of malicious-node-detection models at the fog layer of the proposed BFL-SDWANTrust framework within a multi-controller SD-WAN environment. The input to the algorithm includes a set of controllers $C=\{C_{1},C_{2},\ldots ,C_{n}\}$, where each controller $C_{i}$ collects east–west traffic data at discrete time steps *t*. These data are represented as $D_{ctrl}^{t}=\left\{d_{1}^{t},d_{2}^{t},\ldots ,d_{n}^{t}\right\}$, where each $d_{k}^{t}$ is a feature vector corresponding to an observed traffic flow at time *t*. The local training process for each controller involves three model types: a general local model $T_{i}^{t}$, a convolutional neural network (CNN)-based model $T_{\mathrm{CNN},i}^{t}$ and a support vector machine (SVM)-based model $T_{\mathrm{SVM},i}^{t}$. The learning rate μ controls the step size for gradient descent updates, while *B* denotes the batch size used during CNN training and λ is the regularization parameter applied to the SVM updates to avoid overfitting. At each iteration, the algorithm first computes the gradient $\nabla L(T_{i}^{t},D_{ctrl}^{t})$ of the loss function L over the local dataset, which quantifies the error between the model predictions and the true labels. Using this gradient, the general local model is updated by taking a step proportional to the learning rate and gradient magnitude, following the rule $T_{i}^{t+1}=T_{i}^{t}-\mu \cdot \nabla L\left(T_{i}^{t},D_{ctrl}^{t}\right)$. For the CNN model, training proceeds in batches: for each batch of samples $\left(x_{j},y_{j}\right)$, the gradient of the CNN-specific loss $L_{\mathrm{CNN}}$ is computed and aggregated across all batches. The CNN model is then updated by subtracting the average batch gradient scaled by the learning rate, as expressed by $T_{\mathrm{CNN},i}^{t+1}=T_{\mathrm{CNN},i}^{t}-\mu \cdot \frac{1}{B}\sum_{j=1}^{B}\nabla L_{\mathrm{CNN}}\left(f_{\theta }\left(x_{j}\right),y_{j}\right)$, where $f_{\theta }$ represents the CNN function parameterized by weights θ. The SVM model update involves minimizing the hinge loss, which aims to maximize the margin between classes. For each sample, the algorithm evaluates the indicator function $I(y_{j}\cdot \langle T_{\mathrm{SVM},i}^{t},x_{j}\rangle <1)$, where 〈·,·〉 denotes the dot product between the model weights and input vector and $y_{j}\in \left\{-1,1\right\}$ is the binary class label. The update rule combines a regularization term $\lambda \cdot T_{\mathrm{SVM},i}^{t}$ with a correction term proportional to misclassified samples, following $T_{\mathrm{SVM},i}^{t+1}=T_{\mathrm{SVM},i}^{t}-\mu \cdot \sum_{j}\left(\lambda \cdot T_{\mathrm{SVM},i}^{t}-y_{j}x_{j}\cdot I\left(\cdots \right)\right)$ [55].

Once the local models have been updated, security and trust are ensured via digital signatures. Each controller $C_{i}$ signs its updated general model $T_{i}^{t+1}$ using its private key $\kappa_{priv}^{i}$, producing a signature $\Sigma_{i}=\mathrm{Sign}\left(T_{i}^{t+1},\kappa_{priv}^{i}\right)$. This cryptographic operation guarantees the authenticity and integrity of the model updates before transmission to the blockchain layer for global aggregation. Upon receiving updates, the blockchain nodes verify signatures using the corresponding public keys $\kappa_{pub}^{i}$ through the validation function $\mathrm{Validate}(\Sigma_{i},\kappa_{pub}^{i})$. If the signature verification succeeds, the model update is accepted and incorporated into the global model aggregation process; otherwise, it is rejected and flagged for security review. This mechanism protects against malicious or unauthorized controllers submitting false or corrupted model updates. Overall, the algorithm integrates federated learning with secure cryptographic operations to enable privacy-preserving, distributed training of malicious-node-detection models, maintaining robustness and trust across the SD-WAN controllers while respecting data locality and confidentiality.

### 3.4. Cloud Aggregation Layer in SD-WAN Controller Plane

The fourth layer of our proposed BFL-SDWANTrust architecture is the cloud layer, which is responsible for securely collecting and integrating the local trust models trained by each distributed SD-WAN controller node. After local model training, each virtual machine encrypts its malicious-node-detection model and securely transmits it to the cloud layer, which helps in ensuring end-to-end confidentiality and integrity, as shown in Equation (Equation 19).(19)$$C_{i}=\mathrm{Encrypt}\left(T_{i},K_{i}\right)$$
where, $T_{i}$ represents the locally trained trust model of controller $C_{i}$ and $K_{i}$ is its encryption key. This encryption mechanism ensures that model updates cannot be accessed by unauthorized entities in the network, which solves the issue of man in a middle attack. On the other hand, the cloud server receives encrypted model updates and decrypts each locally trained model with the decryption key $K_{i}^{-1}$, which ensures effective and reliable verification of digital signatures. After the verification step, the cloud server integrates all the locally trained models and generates a unified global model. In this step, the proposed model utilizes the capabilities of the federated averaging process on all verified and authenticated local models for generating a unified global malicious-node-detection model. The averaging process is performed based on control-plane events and east–west traffic logs used in the training of each local model. The aggregated global model $T_{global}^{t+1}$ is computed according to Equation (Equation 20).(20)$$T_{global}^{t+1}=\sum_{i=1}^{N}\frac{v_{i}}{V}\cdot T_{i}^{t}$$
where, $v_{i}$ is the volume of valid control-plane telemetry entries used in training by controller $C_{i}$, $V=\sum_{i=1}^{N}v_{i}$ is the total volume across all nodes and $T_{i}^{t}$ is the local model from node *i* at aggregation round *t*. This weighted aggregation ensures that nodes with more participation have a high influence in the aggregation of the unified model, which ensures the fairness of the model aggregation process. Furthermore, our proposed model efficiently identifies compromised updates by analyzing deviation patterns from the expected score gradients. If any locally trained model $T_{i}$ deviates from the average model gradient $\bar{g}$ and exceeds the predefined threshold θ, then it is considered to be malicious node, as represented in Equation (Equation 21).(21)$$∥T_{i}-\bar{g}∥>\theta \Rightarrow T_{i}\;\mathrm{flagged}\;\mathrm{as}\;\mathrm{malicious}$$
where, θ is a hyperparameter that is derived via grid search on validation telemetry datasets. Its optimal value is 1.2×σ, where σ is the standard deviation of all trust gradient vectors. The models that have deviations below this threshold are considered to be legitimate, while models with higher deviations are considered to be faculty models. In this way, our proposed model ensures that only authentic and legitimate local models can take part in the global malicious-node-detection model. Once the global model is successfully aggregated, it is encrypted and sent back to all participating controller VMs. Each VM then integrates the global malicious-node-detection knowledge into its local decision-making process, which helps in real time, distributed, and secure malicious-node detection in east–west communication. The entire process of model aggregation at the cloud layer is described in Algorithm 4.

Algorithm 4 operates within the SD-WAN controller plane to securely aggregate locally trained trust models received from distributed SDN controllers. The set of controllers is denoted by $C=\{C_{1},C_{2},\ldots ,C_{N}\}$, where each controller $C_{i}$ trains a local trust model represented by $T_{i}$. These local models are encrypted into ciphertexts $C_{i}=\mathrm{Encrypt}\left(T_{i},K_{i}\right)$ using the encryption key $K_{i}$ associated with each controller, ensuring confidentiality against unauthorized access or man-in-the-middle attacks. The cloud layer receives these encrypted models and utilizes the corresponding decryption keys $K_{i}^{-1}$ to decrypt and retrieve the original trust models ${\hat{T}}_{i}=\mathrm{Decrypt}\left(C_{i},K_{i}^{-1}\right)$. Before inclusion in the global aggregation process, each decrypted model undergoes signature verification to authenticate the source and ensure data integrity. Models that fail verification are discarded, and their respective controllers are flagged for potential malicious activity. The telemetry volume $v_{i}$ reflects the quantity of valid control-plane data entries used by each controller $C_{i}$ for training its local model. The total telemetry volume across all controllers is computed as $V=\sum_{i=1}^{N}v_{i}$, which serves as the normalization factor for weighted aggregation. Verified local models and their corresponding volumes are collected into sets V and W, respectively.
**Algorithm 4** Cloud aggregation layer algorithm for global trust model aggregation**Require:** Set of controllers $C=\{C_{1},\ldots ,C_{N}\}$, encrypted local models $\left\{C_{i}\right\}$, encryption   keys $\left\{K_{i}\right\}$, decryption keys $\left\{K_{i}^{-1}\right\}$, threshold θ, telemetry volumes $\left\{v_{i}\right\}$**Ensure:** Encrypted global trust models $\left\{C_{global}^{i}\right\}$  1: Compute total volume $V\leftarrow \sum_{i=1}^{N}v_{i}$  2: Initialize verified models V←∅, volumes W←∅  3: **for **i=1 to *N* **do**  4:     Receive encrypted model $C_{i}$  5:     Decrypt ${\hat{T}}_{i}\leftarrow \mathrm{Decrypt}\left(C_{i},K_{i}^{-1}\right)$  6:     **if** Signature verification of ${\hat{T}}_{i}$ succeeds **then**  7:         Add ${\hat{T}}_{i}$ to V  8:         Add $v_{i}$ to W  9:     **else**  10:         Discard ${\hat{T}}_{i}$ and flag $C_{i}$  11:     **end if**  12: **end for**  13: Compute weighted average gradient:$$\bar{g}\leftarrow \frac{1}{\sum w_{j}}\sum_{j}w_{j}\cdot {\hat{T}}_{j},\;{\hat{T}}_{j}\in V,w_{j}\in W$$  14: **for** each ${\hat{T}}_{i}\in V$ **do**  15:     Compute deviation $d_{i}\leftarrow ∥{\hat{T}}_{i}-\bar{g}∥$  16:     **if** $d_{i}>\theta$ **then**  17:         Flag ${\hat{T}}_{i}$ as malicious  18:         Remove ${\hat{T}}_{i}$ and corresponding $w_{i}$ from V,W  19:     **end if**  20: **end for**  21: Recompute total volume $V\leftarrow \sum w_{j}$  22: Aggregate global model:$$T_{global}^{t+1}\leftarrow \sum_{j}\frac{w_{j}}{V}\cdot {\hat{T}}_{j}$$  23: **for** i=1 to *N* **do**  24:     Encrypt global model for controller *i*:$$C_{global}^{i}\leftarrow \mathrm{Encrypt}\left(T_{global}^{t+1},K_{i}\right)$$  25:     Transmit $C_{global}^{i}$ to $C_{i}$  26: **end for**  27: **return**
$\left\{C_{global}^{i}\right\}$

The aggregation process begins by calculating the weighted average gradient $\bar{g}$, which represents the average model update across controllers and is computed as $\bar{g}=\frac{1}{\sum_{j}w_{j}}\sum_{j}w_{j}{\hat{T}}_{j}$, where ${\hat{T}}_{j}\in V$ and weights $w_{j}\in W$ correspond to telemetry volumes. This weighted approach ensures that controllers with larger training data have a proportionally greater impact on the global model, enhancing fairness and robustness. Next, the algorithm identifies malicious or faulty local models by measuring the deviation $d_{i}=∥{\hat{T}}_{i}-\bar{g}∥$ of each local model from the average gradient. If the deviation exceeds a predefined threshold θ, determined by prior empirical tuning and typically set to a multiple of the standard deviation of model gradients, the local model ${\hat{T}}_{i}$ is flagged as malicious and excluded from further aggregation. Following the removal of anomalous models, the total volume *V* is recalculated over the remaining valid models and a final weighted aggregation is performed to generate the updated global trust model $T_{global}^{t+1}=\sum_{j}\frac{w_{j}}{V}{\hat{T}}_{j}$. The global model is then encrypted individually for each controller using their respective encryption keys $K_{i}$ to produce $C_{global}^{i}=\mathrm{Encrypt}\left(T_{global}^{t+1},K_{i}\right)$, ensuring confidentiality in transmission back to controllers. By securely distributing the global model updates, this algorithm facilitates a trusted, distributed, and collaborative malicious-node-detection system across the SD-WAN east–west interfaces. Algorithm 4 performs cryptographic operations at each layer, sensing, fog, blockchain, and cloud to ensure the authenticity and integrity of control data exchanged between controllers. While these layers perform similar operations like hashing and signing, their purposes differ. At the sensing and fog layers, signatures ensure that telemetry and detection data originate from a valid controller. At the blockchain layer, signatures authenticate model updates before committing to the ledger, while at the cloud layer, they validate model aggregation results. This stratified security model ensures resilience against replay, injection, and impersonation attacks across the trust architecture. Furthermore, we use selective signature verification using smart contract triggers to reduce cryptographic redundancy, which ultimately ensures that only differential updates or trust-critical communications are signed and validated. Instead of verifying every packet or message, the smart contracts monitor metadata fields such as timestamp, sender identity, and semantic difference with the last committed state to determine whether revalidation is necessary. This dynamic triggering mechanism eliminates unnecessary digital signature generation and validation operations during routine or low-risk controller interactions. Furthermore, hash chaining is employed instead of full rehashing where applicable; by linking only modified blocks or deltas with the preceding hash, we reduce redundant hashing computations while preserving data integrity and traceability across the blockchain and IPFS layers. These techniques are particularly effective in environments with frequent east–west synchronization events, where unchanged control states would otherwise undergo full cryptographic processing. By adopting this optimization strategy, the system significantly reduces processing latency and resource utilization without compromising the security model. These optimizations ensure a balance between layered security and performance efficiency, ensuring the system remains scalable under varying telemetry loads and adaptable to real-time operational requirements.

## 4. Experimental Results and Analysis

To evaluate the effectiveness of the proposed BFL-SDWANTrust model, we utilized the publicly available InSDN dataset, which emulates real-world SDN scenarios involving various types of attacks and benign traffic across controller communication channels [56]. The dataset includes telemetry traces, control messages, and inter-controller coordination logs, allowing for a realistic emulation of east–west traffic patterns in multi-domain SD-WAN environments. To align with our architecture, we applied a preprocessing pipeline that extracted relevant features such as packet delivery ratio, control message anomalies, latency fluctuations, and behavioral divergence, which are critical for trust evaluation. Furthermore, to simulate diverse and evolving network conditions, we introduced controlled variations in node behavior such as partial policy compliance, delayed synchronization, and replayed update messages using a network emulator. These modifications enriched the dataset’s heterogeneity, enabling robust training of the federated detection models and allowing for the evaluation of our system under both normal and adversarial operational dynamics. However, the InSDN dataset was originally constructed using a single SDN controller environment and lacks explicit east–west controller communication flows; we employ it to simulate inter-controller malicious node behaviors using distributed virtual environments. In our extended simulation environment, we replicate the east–west communication by deploying multiple logically independent ONOS controller instances, each with virtualized interlinks mimicking the east–west plane. We utilize these setups to synthesize inter-controller communication logs from the original telemetry by assigning directional flows, configuration exchanges, and behavior consistency patterns. Though the dataset does not natively provide east–west interactions, our framework reinterprets and restructures the control-plane behaviors to evaluate the trustworthiness and detection of potentially malicious east–west transactions, consistent with real-world SD-WAN deployments that extend single-controller telemetry into logically federated controller architectures. The main objective of our model is to ensure effective and reliable malicious-node detection in multi-controller SD-WAN environments while simultaneously preserving the privacy of network entities. We evaluate the performance of our proposed model while considering the factors of accuracy, precision, recall, F1-score, Area Under the Curve (AUC), PDR, average latency, transaction latency, training time, testing time, and blockchain gas consumption. We compare our proposed model with LR, SVM, RF, KNN, XGBoost [7], and DBN [8]. Table 3 provides the simulation parameters and performance results of our proposed model.

### 4.1. Dataset Details

The InSDN Dataset is designed to support research on security and anomaly detection in SDN-WANs, especially in scenarios involving east–west communication between controllers [56]. It is created using Mininet and POX controllers and includes around 70,000 records with both normal and attack traffic. Each record contains important features such as control packet types (e.g., packet_in, flow_mod), source and destination IDs, timestamps, and flow rule actions. In our proposed model, we use this dataset in a federated learning environment where each controller trains its local model using InSDN telemetry. This allows the system to detect abnormal controller behavior without sharing raw data, preserving privacy and reducing bandwidth. The combination of controller-level control-plane data and attack scenarios makes InSDN a strong fit for testing our trust framework for east–west communication security. There are various flow-level network statistics, device logs, and system metrics in the dataset that make it effective and reliable for the evaluation of intrusion detection and malicious-node-detection models in SD-WANs. The dataset captures various heterogeneous network traffic patterns. These patterns come from multiple SDN controllers and edge devices. This helps reflect real-world deployment challenges. The traffic flows are often noisy, imbalanced, and dynamic. This dataset is suitable for a federated-learning-based malicious-node-detection mechanism, where sensitive and important telemetry data remains localized within each SDN controller domain to ensure privacy preservation and effective model training. In our proposed model, we simulate a federated learning mechanism over various SDN controllers, where local model updates are securely aggregated without sending any raw telemetry data outside the network.

### 4.2. Experimental Results

Figure 4 presents the comparison of our proposed BFL-SDWANTrust framework and various machine learning algorithms in terms of accuracy, precision, recall, and F1-score. The figure shows that our proposed model outperforms all benchmark schemes in all four performance metrics, which shows that our model is effective and reliable for secure east–west communication in multi-controller SD-WAN environments. The accuracy of our proposed model is 98.8%, which is higher than all other classifiers, as shown in Table 4. This is because our proposed model utilizes the capabilities of blockchain and federated learning. The federated learning enables decentralized training across distributed SD-WAN nodes, which preserves the privacy of data while simultaneously learning localized trust behaviors. On the other hand, the blockchain distributed network efficiently ensures data integrity and immutability, which further enhances the accuracy and reliability of the model. Besides this, the precision of our proposed model is 98.0%, which shows our model has a very low false positive rate. Our model has better precision than all other classifiers, which have accuracies of 89.4% and 88.9%, respectively. The reason for the low accuracy of these models is that these classifiers are vulnerable to various issues, such as outliers and static decision boundaries. Our proposed model is very effective and reliable for malicious-node detection in SD-WANs because misclassification of legitimate controllers as malicious nodes can affect the overall performance of the model.

Furthermore, Figure 4 shows that our model has the highest recall value of 97.0%, which shows that our model has a better capability to identify actual malicious behaviors. This is the federated learning mechanism and the blockchain-based decentralized consensus algorithms in which malicious-node-detection models are aggregated from multiple nodes, which reduces bias and enhances threat coverage. Lastly, our proposed model achieves the highest F1-score of 97.5%. The reason for the low F1-score of other classifiers is that these models are not scalable and adaptive in dynamic SD-WAN topologies. In this way, our proposed model is not only accurate but also more precise, sensitive, and resilient for malicious-node-detection in SD-WANs. Furthermore, the blockchain architecture makes it very suitable for scalable, real-time malicious-node-detection in multi-controller SD-WANs.

Figure 5 provides the confusion matrix for the proposed BFL-SDWANTrust model on the InSDN dataset for Scenario 1. The matrix evaluates the classification performance of our proposed model across four distinct traffic categories: normal, DDoS, recon, and backdoor. The results show that our model has high accuracy and strong classification capabilities. Furthermore, the results show that our model efficiently identified 240 out of 243 normal entities, with only 2 misclassified as DDoS attackers and 1 as a recon attacker. For the DDoS class, 230 out of 238 entities were correctly classified, with 3 misclassifications as normal entities, 4 recon attackers, and 1 backdoor attacker, as shown in Table 5. The reason for these minor errors is the overlapping of packet-level attributes with legitimate traffic. Furthermore, it can be easily predicted that the recon attacker class has a strong classification performance, with 225 true positives out of 230. Only 2 instances are misclassified as normal entities and 3 as backdoor attackers, which shows that our model is efficient and reliable in identifying malicious nodes. Lastly, the model efficiently classifies backdoor attacker samples, as 216 samples are correctly identified out of 219, with 1 misclassification as normal entities and 2 as DDoS attackers. Overall, the results show that our proposed model has high precision and recall across all classes, making the model effective and reliable in classifying various types of attackers while simultaneously preserving the privacy of network entities.

Figure 6 shows the confusion matrix of the proposed BFL-SDWANTrust model for Scenario 2 on the InSDN dataset. There are five classes in this scenario: normal, DDoS, recon, backdoor, and injection, which show a reliable entity in the network. The results show that our model correctly predicts 190 normal entities out of 194 and misclassifies only 3 as DDoS attackers and 1 as an injection attacker, as shown in Table 6. This shows that our proposed model is effective and reliable for malicious-node-detection for all these node types. In the DDoS category class, 180 samples are correctly identified; however, only 5 are misclassified, which shows that our model has a high recall value in the detection of various attacks. Besides this, the recon class has 175 correct classifications from 180 samples. However, the class overlaps with backdoor and injection, which is due to packet-level similarities and common patterns in both attacks. Furthermore, the backdoor class has 170 correct samples out of 174. However, there are a few misclassifications, which show that the backdoor traffic has some resemblance to other attack patterns in SD-WAN environments. Lastly, the injection class has effective and reliable results with 180 true positives out of 184, which shows that our fine-grained model is efficiently able to identify various attacks on the application layer. The results show that our model effectively classifies almost all legitimate and malicious nodes in the network. The reason is that in our model, the models are locally trained at edge nodes with class-balanced data. Due to this, our model has high accuracy, reduced poisoning, and data privacy, which makes BFL-SDWANTrust a robust and reliable solution for secure malicious-node-detection in east–west SD-WANs.

Figure 7 provides the PDR comparison of our proposed BFL-SDWANTrust model and six benchmark models: LR, SVM, RF, KNN, XGBoost, and DBN. The figure shows that our proposed model has the lowest PDR of 0.8% and outperforms all models. The reason is that our model efficiently integrates blockchain and a federated learning mechanism. The blockchain ensures data immutability while simultaneously ensuring reliable and trusted data transmission across east–west SD-WAN controllers. On the other hand, the distributed federated learning not only reduces transmission overhead local trained model but also preserves the privacy of users. KNN has the highest PDR of 3.5%, which is higher than LR (3.2%) and SVM (2.9%). The reason is that KNN is efficiently able to accurately classify malicious and legitimate nodes based on their local patterns. Furthermore, our proposed model efficiently minimizes network congestion and ensures that there is no redundant packet forwarding in the network, which leads to fewer dropped packets. The results in Table 7 show the reliability and efficiency of our proposed BFL-SDWANTrust model in handling traffic in complex SD-WANs.

Figure 8 provides the latency comparison of our proposed BFL-SDWANTrust model and six benchmark models: LR, SVM, RF, KNN, XGBoost, and DBN. The results shows that our proposed model has lowest latency of 12.4 ms and outperforms all benchmark classification models: KNN (39.8 ms), LR (35.6 ms), SVM (32.1 ms), RF (28.7 ms), DBN (23.9 ms) and XGBoost (27.5 ms), as shown in Table 8 because our proposed model utilizes the capabilities of blockchain and federated learning that ensure quick and reliable malicious-node detection across SD-WAN controllers. Furthermore, our model solves the issues of performance bottlenecks as there is no involvement of any centralized or third party in our proposed model. Besides this, we also use lightweight locally trained models in the east–west communication layer, which helps in faster packet handling and reduces end-to-end delay. Hence, our proposed BFL-SDWANTrust model is efficient and reliable for real-time malicious-node-detection in SD-WANs.

Figure 9 shows the comparison of our proposed BFL-SDWANTrust model and benchmark schemes in terms of precision–recall (PR) curves. Our model achieves an AUC of 0.998, which shows that our model has high classification performance and robustness against class imbalance. The DBN model has an AUC of 0.932, while XGBoost and RF achieve AUCs of 0.915 and 0.907, respectively because DBN can learn hierarchical representations, which helps in better feature extraction and capturing complex attack patterns. On the other hand, SVM (0.893), LR (0.881), and KNN (0.874) have low AUC values, as shown in Table 9 because our models are not able to efficiently handle complex patterns of various attacker nodes in SD-WANs. The high AUC value indicates that our model has a high true positive rate and low false positives due to the effective and reliable integration of federated learning and blockchain trust mechanisms. In this way, our model not only achieves high malicious-node-detection accuracy but also ensures secure and privacy-preserving east–west controller communication in multi-controller SD-WANs.

Figure 10 provides a comparative analysis of gas consumption in Gwei across ten virtual machines (VMs) within the SD-WANs (VM-1 to VM-10) in terms of PoW (23,000–25,700 Gwei) and PoA (22,500–25,500 Gwei) consensus algorithms. The figures show that PoW requires higher gas usage compared to PoA across all 10 VMs. The figure shows that VM-4 and VM-9 have the highest gas consumption for PoW (25,700 Gwei), while PoA has lower gas consumption (25,500 Gwei). Furthermore, the average consumption across all VMs for PoW is around 24,790 Gwei, whereas PoA averages around 24,250 Gwei, as depicted in Table 10. The reason is that PoW selects a miner VM based on its computational capabilities. This miner node is responsible for validating the transaction across network entities and adding the blocks into the blockchain. This miner VM is selected through a competition among all interested VMs; each VM needs to solve a computational puzzle, and only one VM is selected as the miner HU. The resources of all other participants are wasted. On the other hand, the PoA consensus algorithm ensures that only trusted and pre-authorized SDN controllers can validate east–west communication transactions and add the blocks to the distributed blockchain ledger. Each interested SDN controller shows its interest in becoming a validator for the current transactions. However, only controllers with a high reputation and consistent historical behavior are selected for transaction validation and adding the block without the utilization of large network computational resources, as compared to the traditional PoW consensus algorithm. It is the reason that PoA is a more scalable and reliable consensus algorithm for inter-controller communication in the SD-WAN malicious-node-detection mechanism.

Figure 11 compares the transaction latency of two cryptographic hashing algorithms, Keccak-256 and SHA-256, for various transactions in our proposed BFL-SDNWanTrust model. The latency for both algorithms increases linearly with the number of transactions, which shows that the computational overhead is associated with the processing of transactions with a larger workload. In our proposed model, we utilize the capabilities of the Keccak-256 hashing algorithm for ensuring the security and integrity of data, in contrast with the benchmark schemes [7,8] that use SHA-256. The results show that Keccak-256 has lower transaction latency as compared to SHA-256 for all transactions. For instance, Keccak-256 has a latency of approximately 225 s at 13,000 transactions while SHA-256 has a higher latency of around 250 s, as shown in Table 11. The reason is that the Keccak-256 hashing algorithm uses sponge construction, which takes a very small amount of time to hash as compared to the SHA-256 Markel tree technique. This latency difference shows that Keccak-256 is efficient and reliable in quick and intelligent decision-making in the blockchain network. In this way, our proposed model utilizes the capabilities of blockchain and ensures the optimization of malicious-node-detection mechanisms in SD-WANs.

Figure 12 and Figure 13 show a comparison of our proposed model and benchmark schemes in terms of training times and testing times, respectively. Figure 12 shows that our proposed model achieves the shortest training time of 12 s. This is faster than other models, such as XGBoost and DBN, which take over 40 s to train the model, as shown in Table 12 because our proposed model uses a lightweight architecture and a decentralized learning mechanism, which minimizes redundant computations. Furthermore, the federated learning mechanism reduces data transmission overhead, which ensures faster convergence and scalable training across distributed SD-WAN nodes. In this way, our proposed BFL-SDWANTrust model is effective and reliable for real-time malicious-node detection in SD-WANs. On the other hand, Figure 13 shows that our proposed BFL-SDWANTrust model has the most efficient performance with the lowest testing time of 3.1 s, as shown in Table 12. The reason for this is that our proposed decentralized architecture of BFL-SDWANTrust enables parallel computation and model training across federated nodes. Furthermore, our model has a lightweight design, which minimizes computational latency during a real-time malicious-node-detection mechanism. This reduced testing time shows that our proposed BFL-SDWANTrust model is highly suitable for real-time malicious detection mechanisms where low latency is very significant.

Figure 14 shows a comparative analysis of key performance metrics such as latency, throughput, and accuracy between scenarios with and without cryptographic modules in the BFL-SDWANTrust framework. The figure shows that the cryptographic operations, such as digital signatures, hashing, and verification mechanisms, increase latency from 80 ms to 90.16 ms, representing a 12.7% rise. This is due to the additional computation time required for encrypting east–west controller messages, verifying identities through smart contracts, and hashing control packets stored in IPFS. On the other hand, throughput experiences a reduction from 120 Mbps to 110 Mbps (an 8.3% decrease), primarily due to CPU contention and queuing delays introduced by cryptographic tasks. These include real-time signature generation, validation, and hash recalculations across multilayer controller communications, which consume compute resources and reduce packet processing speed.

Besides this, the accuracy improves significantly from 85.0% to 98.8%. This improvement is due to the integration of hybrid CNN+SVM classification in the federated learning pipeline. The cryptographic mechanisms enhance data integrity and authenticity, resulting in more reliable training data and better global model convergence. Signature verification acts as a filtering layer, discarding poisoned or manipulated telemetry data before aggregation, thus boosting the detection accuracy of malicious controllers. The trade-off shown in this plot highlights a well-justified balance between security and performance. The minor overheads in latency and throughput are acceptable in light of the substantial gain in accuracy, proving the feasibility of secure, real-time east–west trust validation in SD-WAN environments (Table 13).

## 5. Discussion

The BFL-SDWANTrust framework integrates federated learning and blockchain technologies to establish a privacy-preserving trust evaluation mechanism specifically tailored for multi-controller SD-WAN environments. This design ensures that sensitive data are kept local to each domain while enabling collaborative learning across distributed controllers. The layered architecture of BFL-SDWANTrust introduces inherent robustness, decentralization, and scalability by distributing tasks across the edge, fog, and cloud layers. However, this layered approach also brings certain trade-offs in terms of latency and computational overhead, particularly due to the cryptographic operations such as digital signatures, hash chaining, and data encryption applied at multiple stages. These security mechanisms significantly enhance data integrity and trustworthiness, albeit with increased resource consumption, especially in the fog and cloud layers where aggregation and verification tasks are computationally intensive.

Despite these overheads, empirical evaluations show that the trade-offs are well justified by the substantial improvements in malicious-node-detection accuracy, precision, and recall. The system achieves this through the deployment of a hybrid CNN+SVM classifier, which combines the deep feature extraction capabilities of CNNs with the discriminative properties of SVMs for precise threat detection. The classifier is locally trained at each controller and validated collaboratively through federated aggregation, ensuring both performance and data privacy. To address the latency issues commonly associated with blockchain, the framework employs a PoA consensus mechanism, which eliminates the need for energy-intensive mining processes and reduces block confirmation times. This makes the system lightweight, scalable, and suitable for real-time SD-WAN operations. Additionally, the integration of IPFS for distributed off-chain storage significantly reduces the storage burden on the blockchain ledger, allowing efficient management of logs, models, and audit trails without bloating the chain. The BFL-SDWANTrust model also emphasizes modularity and adaptability, enabling easy integration into other critical infrastructures such as smart grids, healthcare systems, and intelligent transportation networks, where secure inter-domain communication and trust validation are paramount. Nevertheless, tuning the hybrid detection model across heterogeneous controllers introduces challenges in terms of model convergence, data heterogeneity, and resource synchronization, which need continuous validation and performance monitoring. Overall, BFL-SDWANTrust achieves a balanced integration of security, performance, and scalability, offering a promising direction for the development of secure, intelligent, and collaborative SD-WAN infrastructures.

## 6. Conclusions and Future Work

In this paper, a novel malicious-node-detection model, BFL-SDWANTrust, was proposed that solves the issues of centralized malicious-node-detection schemes and third-party-based multi-controller SD-WANs. The issues of privacy leakage, performance bottlenecks, and single points of failure were solved while utilizing the capabilities of federated learning and blockchain techniques. In our proposed model, the distributed model training was ensured at edge nodes without central data aggregation. This distributed model training not only enhances the accuracy of the malicious-node-detection process but also preserves the privacy of network entities. Furthermore, we utilized the capabilities of blockchain to ensure transparent and tamper-proof validation of node activities and network communication transactions. We compared our proposed model with various benchmark schemes on the InSDN dataset. The results show that BFL-SDWANTrust outperforms all existing benchmark models in terms of accuracy (98.8%), precision (98.0%), recall (97.0%), and F1-score (97.7%). Moreover, our model achieves the lowest training (12 s) and testing times (3.1 s) as compared to other classification models. In this way, our proposed model minimizes network overhead by eliminating centralized entities and ensures effective and reliable malicious-node detection while simultaneously preserving the privacy of network entities in east–west communication in SD-WANs. The proposed framework is applicable in nationwide SD-WAN deployments, where decentralized coordination between controllers is critical. This includes large-scale enterprise networks, internet service provider (ISP) backbones, and cross-domain smart city infrastructures. Additionally, the model is suitable for privacy-sensitive sectors such as healthcare, defense, and finance, where secure trust evaluation and policy consistency are essential. The ability to detect misbehaving controllers in real time while preserving data privacy makes BFL-SDWANTrust suitable for next-generation networked systems operating across distributed environments.

Our model has a large computational overhead due to the integration of blockchain with federated learning, which leads to increased latency and resource consumption in large-scale SD-WANs. Furthermore, the proposed model is not adaptive to concept drift and changes in attack patterns over time, which reduces the accuracy of the detection mechanism in evolving networks. As future work, we propose robust aggregation methods such as Krum or trimmed mean in the federated learning framework to solve the issues of poisoning attacks and improve the trustworthiness of local model updates. Furthermore, we propose the integration of Hyperledger Fabric as a permissioned blockchain layer to improve transaction throughput, reduce energy consumption, and provide fine-grained access control in multi-controller SD-WAN environments. Additionally, adaptive learning mechanisms can also be integrated into the federated model to dynamically handle concept drift and evolving attack vectors, which will ensure effective detection accuracy over time. 

## Figures and Tables

**Figure 1 sensors-25-05188-f001:**
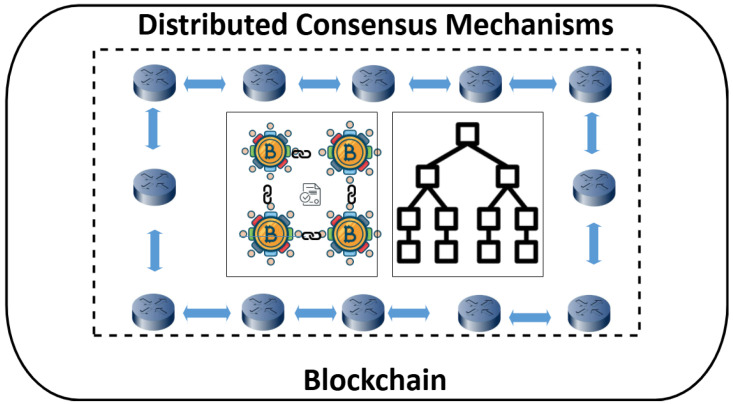
A general blockchain layered architecture.

**Figure 2 sensors-25-05188-f002:**
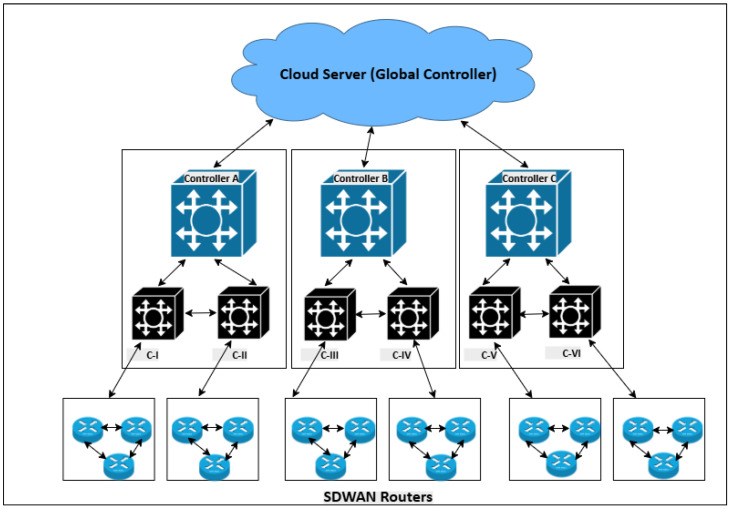
Hierarchical and distributed SDN controller architecture.

**Figure 3 sensors-25-05188-f003:**
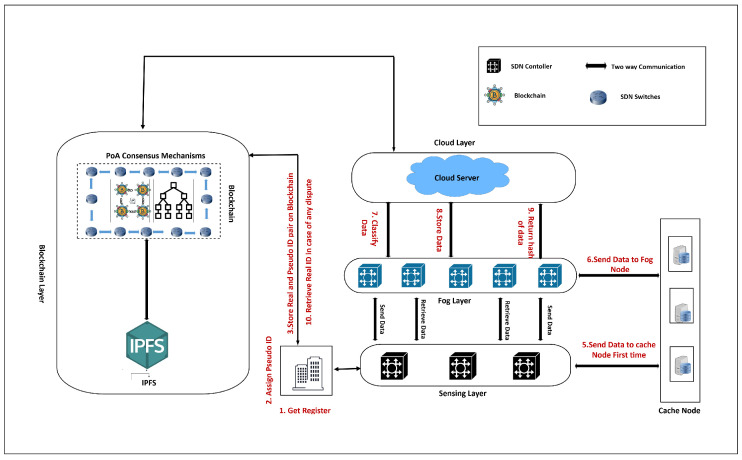
Proposed model for secure east–west communication in multi-controller SD-WANs.

**Figure 4 sensors-25-05188-f004:**
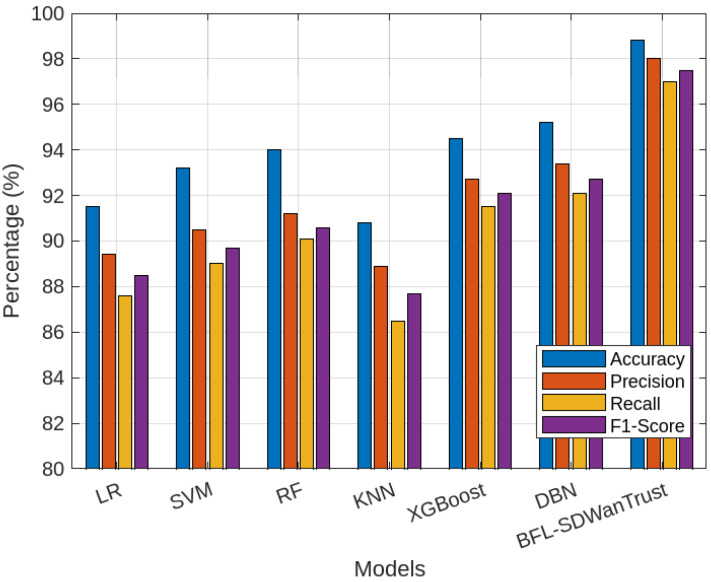
Comparison of various performance metrics among malicious-node-detection models on InSDN dataset.

**Figure 5 sensors-25-05188-f005:**
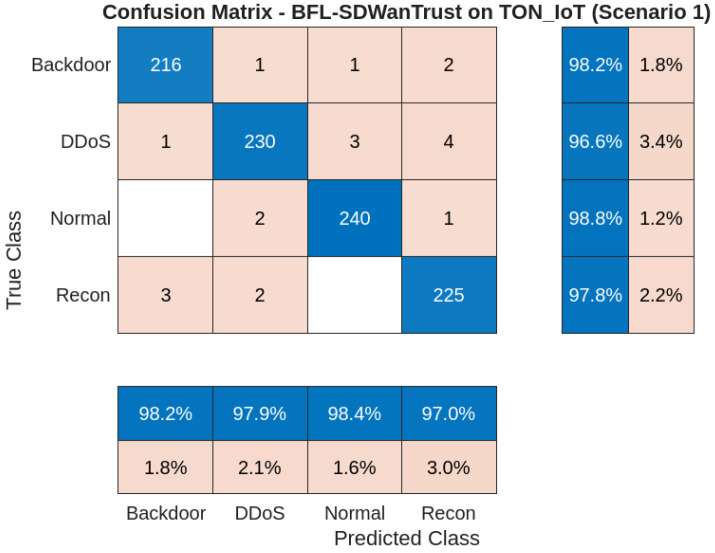
Proposed BFL-SDWANTrust confusion matrix for Scenario 1 malicious-node detection on InSDN dataset.

**Figure 6 sensors-25-05188-f006:**
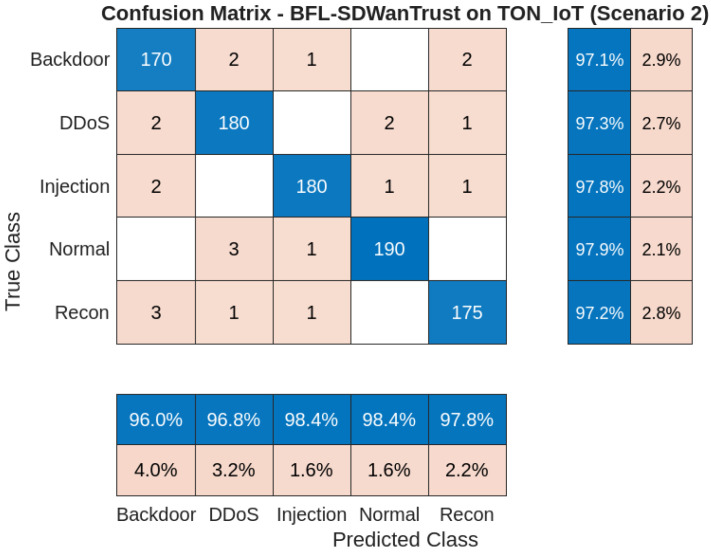
Proposed BFL-SDWANTrust confusion matrix for Scenario 2 malicious-node detection on InSDN dataset.

**Figure 7 sensors-25-05188-f007:**
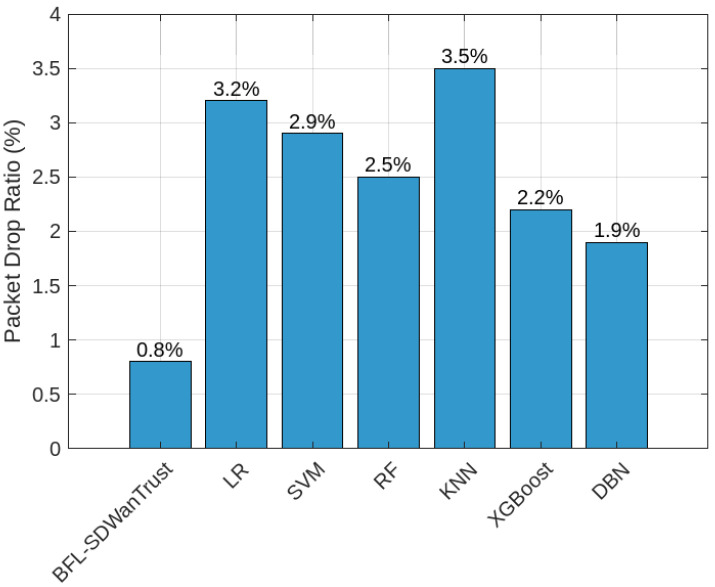
Comparison of PDR among malicious-node-detection models on InSDN dataset.

**Figure 8 sensors-25-05188-f008:**
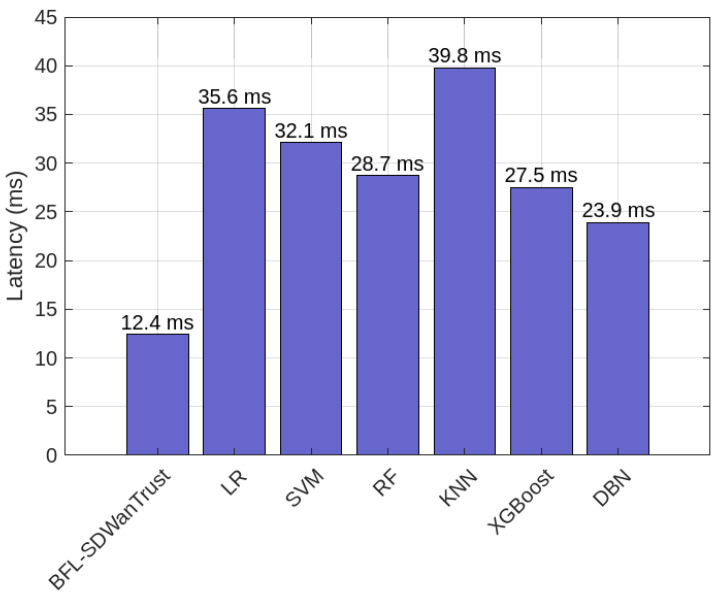
Comparison of latency among malicious-node-detection models on InSDN dataset.

**Figure 9 sensors-25-05188-f009:**
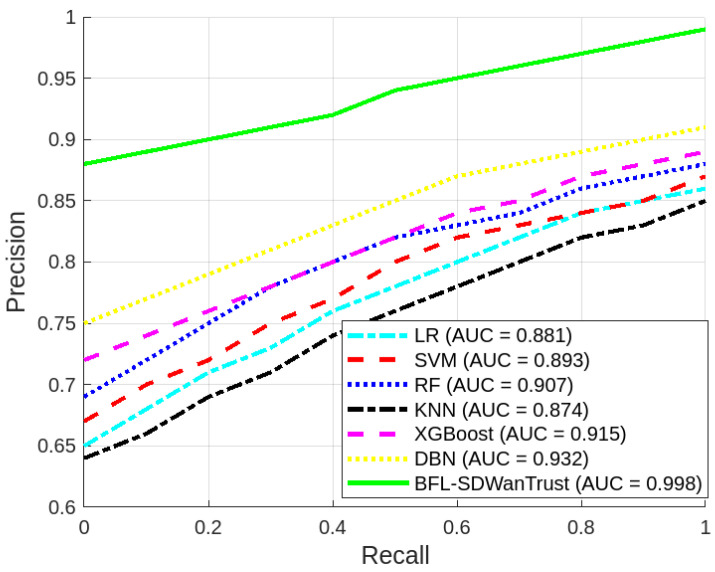
Comparison of AUC among malicious-node-detection models on InSDN dataset.

**Figure 10 sensors-25-05188-f010:**
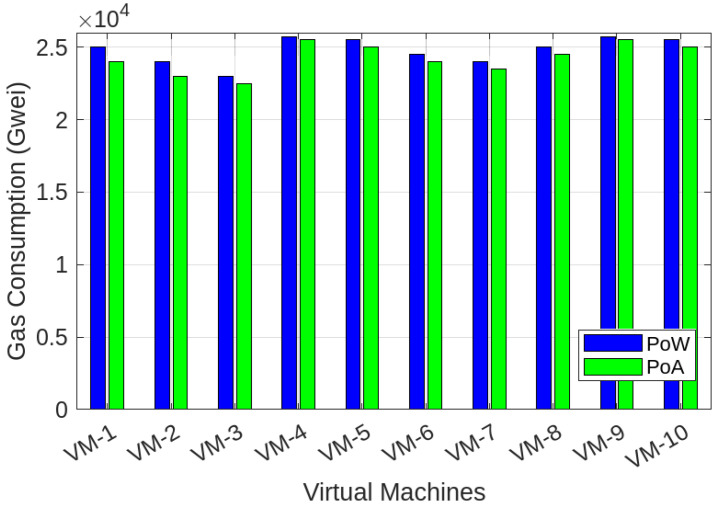
Comparison of consensus algorithms among malicious-node-detection models.

**Figure 11 sensors-25-05188-f011:**
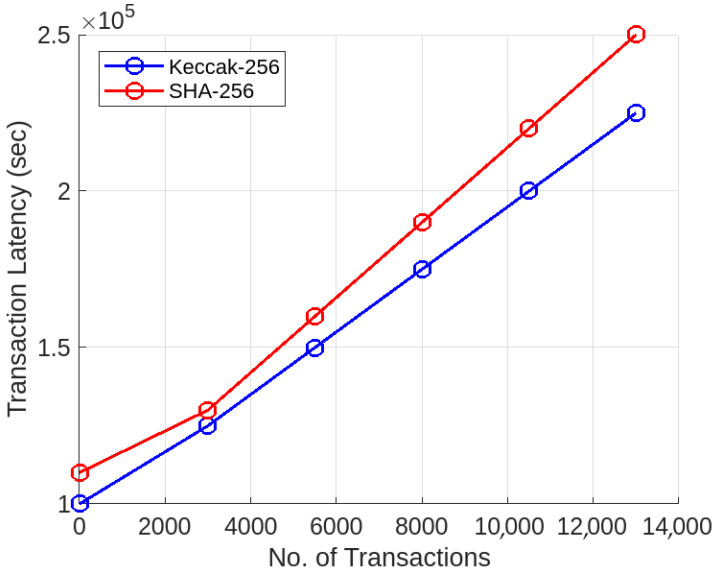
Comparison of transaction latency among malicious-node-detection models.

**Figure 12 sensors-25-05188-f012:**
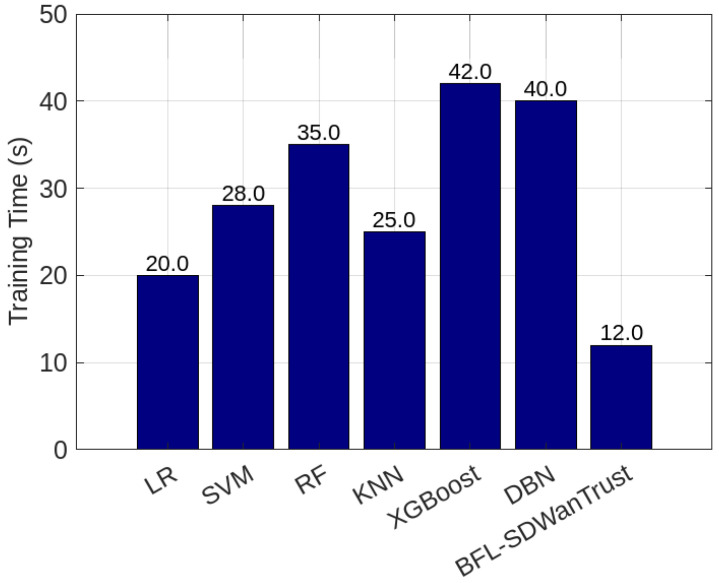
Comparison of training time among malicious-node-detection models.

**Figure 13 sensors-25-05188-f013:**
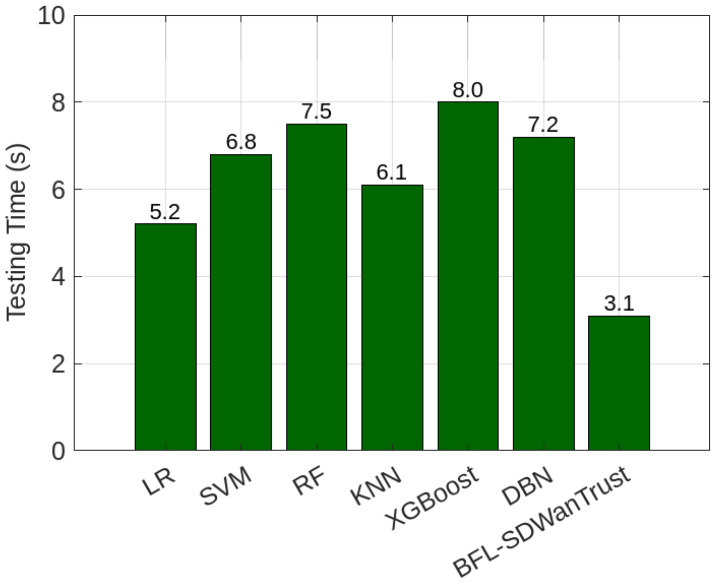
Comparison of testing time among malicious-node-detection models.

**Figure 14 sensors-25-05188-f014:**
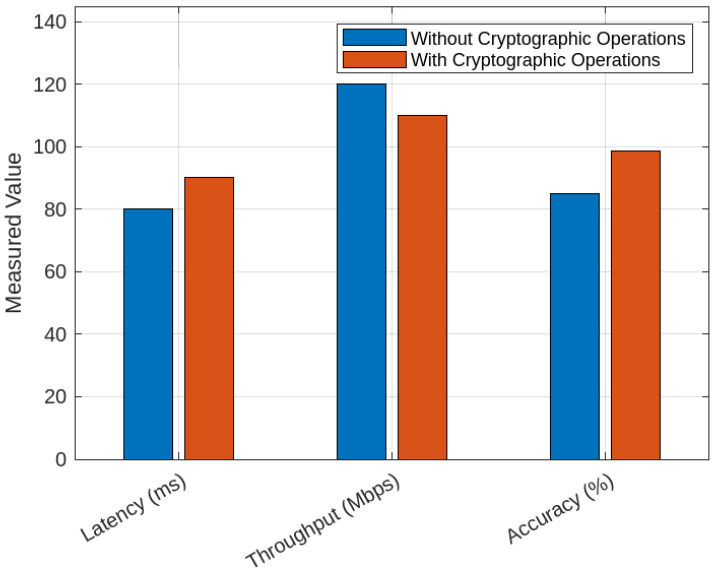
Performance metrics with and without cryptographic operations.

**Table 1 sensors-25-05188-t001:** Abbreviations used in the proposed model.

Abbreviation	Description
AUC	Area Under the Curve
CNN	Convolutional Neural Network
FL	Federated Learning
IPFS	InterPlanetary File System
PDR	Packet Delivery Ratio
PK	Public Key
PoA	Proof of Authority
SDN	Software-Defined Network
SD-WAN	Software-Defined Wide-Area Network
SK	Secret Key
SVM	Support Vector Machine
TLS	Transport Layer Security
VM	Virtual Machine

**Table 2 sensors-25-05188-t002:** Comprehensive analysis of existing approaches: Challenges, innovations, evaluations, and future directions.

Existing Model	Addressed Limitations	Performance Parameters	Research Gaps
A blockchain-based distributed mechanism is proposed that ensures distributed credential-based access control and encrypted security of data while utilizing the capabilities of the Ethereum network and customized blockchain protocols [34].	Existing techniques are not able to secure the east–west interface and communication between SDN nodes, which compromises the security of the overall network.	Authentication time, decentralization, scalability, authentication, system latency, authentication latency, registration latency, throughput, and access control.	The proposed model is highly dependent on the Ethereum network, which leads to large network overhead, such as increased latency and operational costs due to blockchain processing overhead.
A blockchain and federated-learning-enabled security protocol is proposed that efficiently detects denial of services, distributed denial of services, and replay attacks and prevents the network from them [36].	The traditional techniques are not able to efficiently detect and safeguard the network against these attacks. SDN networks are vulnerable to several attacks due to the rapid increase in IoT devices.	Data flow rate, accuracy, true positive rate, recall, true negative rate, and velocity updates.	The proposed model is vulnerable to the issues of performance degradation, as the proposed model has a large computational overhead of federated learning.
A decentralized attribute-based access control mechanism is proposed that removes centralized entities from the network and ensures effective and reliable cross-domain access control [37].	Existing attribute-based access control mechanisms are not able to efficiently ensure security of cross-domain networks due to the involvement of a centralized attribute evaluation party.	Number of predicates, number of attributes, model performance, and execution time.	The proposed model is vulnerable to the issue of performance bottlenecks due to the large computational overhead of group signature schemes, especially when handling many attributes or predicates.
A blockchain-enabled adaptive fading reputation mechanism is proposed	SDN networks are vulnerable to various attacks, such as false data injection attacks	Detection rate, execution time, number of switches, number of hosts	The operations of the proposed model are highly dependent on a master controller of a respective domain.
effective and optimal detection of malicious nodes in the network [38].	SDN networks are vulnerable to various attacks, such as false data injection attacks, denial of service, and distributed denial of service. Due to these attacks, the SDN networks are facing the issue of topology inconsistency among network controllers.	detection time, reputation, and number of observation intervals.	It causes the issue of a single point of failure within each domain, which compromises the decentralization objective of the proposed model.
A machine learning and blockchain-enabled SDN is proposed that ensures network security while simultaneously preserving the privacy of entities and the flexibility of the network [39].	Fifth-generation networks are vulnerable to various issues such as low data protection, entity privacy leakage, and data loss.	Topology, bandwidth, throughput, node failure rate, computational delay, and bandwidth prediction.	The proposed model has a large computational overhead and network latency due to the training of LSTM and MLP classifiers, especially in real-time high-speed fifth-generation networks.

**Table 3 sensors-25-05188-t003:** Simulation parameters and performance metrics for proposed BFL-SDWANTrust using the InSDN dataset.

Parameter Name	Values
Dataset Used	InSDN Dataset (2020)
Number of Network Flows	Over 2 million labeled telemetry records
Network Environment	Multi-controller SD-WAN with east–west communication
Data Features	Telemetry metrics, flow statistics, device logs, network metadata
Data Preprocessing	Normalization, feature encoding, imputation, dimensionality reduction
Federated Learning Setup	10 ONOS SDN controllers as distributed clients
Communication Rounds	150 federated training rounds
Model Architecture	BFL-SDWANTrust: CNN + SVM with Blockchain-based Federated Learning
Optimizer	Adam optimizer (learning rate = 0.001) for CNN; Grid search for SVM
Loss Function	Cross-Entropy Loss (CNN); Hinge Loss (SVM)
Evaluation Metrics	Accuracy, Precision, Recall, F1-Score, AUC, Transaction Latency, Training Time, Testing Time, Packet-Drop Ratio
Accuracy Achieved	99.8% (on test data)
Precision	98.0%
Recall	97.0%
F1-Score	97.5%
AUC Score	0.998
Transaction Latency	13.6 ms (lowest among all compared models)
Training Time	12.0 s (lowest)
Testing Time	3.1 s (lowest)
Packet-Drop Ratio	0.8% (lowest)
Comparison Models	LR (91.5%), SVM (93.2%), RF (94.0%), KNN (90.8%), XGBoost (94.5%), DBN (95.2%), BFL-SDWANTrust (99.8%)
Privacy Preservation	Blockchain-integrated model aggregation with smart contract enforcement
Hardware Used	NVIDIA RTX 4090, 128 GB RAM

**Table 4 sensors-25-05188-t004:** Performance metrics comparison of BFL-SDWANTrust and benchmark models.

Model	Accuracy (%)	Precision (%)	Recall (%)	F1-Score (%)
LR	91.5	89.4	87.6	88.5
SVM	93.2	90.5	89.0	89.7
RF	94.0	91.2	90.1	90.6
KNN	90.8	88.9	86.5	87.7
XGBoost	94.5	92.7	91.5	92.1
DBN	95.2	93.4	92.1	92.7
**BFL-SDWANTrust**	**98.8**	**98.0**	**97.0**	**97.5**

**Table 5 sensors-25-05188-t005:** Confusion matrix for BFL-SDWANTrust on InSDN dataset (Scenario 1).

Actual / Predicted	Normal	DDoS	Recon	Backdoor
**Normal**	240	2	1	0
**DDoS**	3	230	4	1
**Recon**	2	0	225	3
**Backdoor**	1	2	0	216

**Table 6 sensors-25-05188-t006:** Confusion matrix for BFL-SDWANTrust on InSDN dataset (Scenario 2).

Actual / Predicted	Normal	DDoS	Recon	Backdoor	Injection
**Normal**	190	3	0	0	1
**DDoS**	2	180	1	2	0
**Recon**	0	1	175	3	1
**Backdoor**	0	2	2	170	1
**Injection**	1	0	1	2	180

**Table 7 sensors-25-05188-t007:** Packet-drop ratio comparison of BFL-SDWANTrust and baseline models.

Model	Packet-Drop Ratio (%)
BFL-SDWANTrust	**0.8**
LR	3.2
SVM	2.9
RF	2.5
KNN	3.5
XGBoost	2.2
DBN	1.9

**Table 8 sensors-25-05188-t008:** Latency comparison of BFL-SDWANTrust and baseline models.

Model	Latency (ms)
BFL-SDWANTrust	**12.4**
LR	35.6
SVM	32.1
RF	28.7
KNN	39.8
XGBoost	27.5
DBN	23.9

**Table 9 sensors-25-05188-t009:** Area under precision–recall curve (AUC) Comparison.

Model	AUC (PR)
BFL-SDWANTrust	**0.998**
DBN	0.932
XGBoost	0.915
RF	0.907
SVM	0.893
LR	0.881
KNN	0.874

**Table 10 sensors-25-05188-t010:** Gas consumption comparison (PoW vs. PoA).

VM Instance	PoW (Gwei)	PoA (Gwei)
VM-1	25,000	24,000
VM-2	24,000	23,000
VM-3	23,000	22,500
VM-4	25,700	25,500
VM-5	25,500	25,000
VM-6	24,500	24,000
VM-7	24,000	23,500
VM-8	25,000	24,500
VM-9	25,700	25,500
VM-10	25,500	25,000

**Table 11 sensors-25-05188-t011:** Transaction latency comparison of Keccak-256 and SHA-256.

No. of Transactions	Keccak-256 Latency (s)	SHA-256 Latency (s)
1500	100,000	110,000
3000	125,000	130,000
5500	150,000	160,000
8000	175,000	190,000
10,500	200,000	220,000
13,000	225,000	250,000

**Table 12 sensors-25-05188-t012:** Training and testing time comparison of different models.

Model	Training Time (s)	Testing Time (s)
LR	20.0	5.2
SVM	28.0	6.8
RF	35.0	7.5
KNN	25.0	6.1
XGBoost	42.0	8.0
DBN	40.0	7.2
BFL-SDWANTrust	12.0	3.1

**Table 13 sensors-25-05188-t013:** Performance metrics comparison with and without cryptographic modules in BFL-SDWANTrust.

Scenario	Latency (ms)	Throughput (Mbps)	Accuracy (%)
Without Cryptographic Operations	80.00	120.00	85.0
With Cryptographic Operations	90.16	110.00	98.8

## Data Availability

The original contributions presented in this study are included in the article. Further inquiries can be directed to the corresponding author.

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
