# Peer review of "BFL-SDWANTrust: Blockchain Federated-Learning-Enabled Trust Framework for Secure East–West Communication in Multi-Controller SD-WANs"

_sensors, 2025, doi:10.3390/s25165188_

Round 1
Reviewer 1 Report
Comments and Suggestions for Authors
The authors propose a blockchain-FL architecture on SD-WANs. They use blockchains as distributed storage to avoid the single-point-of-failure issue and apply IPFS for the communication storage. They also propose an integration method for multiple learning techniques. However, this work really confuses me and it is hard for me to catch the point. My comments are as follows.
- According to the title, this work is for securing east-west communication in multi-controller. So I think the authors use some east-west communication dataset for machine learning. However, they use TON-IoT dataset which seems irrelative to east-west communication. Instead, the authors try to use this dataset for IoT attacks training. For me, they are two different topics and should not be put in one work.
- The authors claim that only trusted SDN controller can be registered on the blockchain. That is, all SDN controllers are trusted. Note that the communications between controllers undoubtedly are protected through secure channels like TLS. So what kind of attacks are launched on the east-west communication? Frankly speaking, after reading this paper, I have no idea.
- The authors use IPFS for storing the control data. The authors use many descriptions about the signature design. However, I do not know how these data are used to secure the east-west communication.
- The related work needs more descriptions. For example, the authors claim that the model of [21], which is dependent on the Ethereum, leads to large network overhead. However, this work is still based on the blockchain. So what is the difference between this work and [21]? The authors should provide more reasons.
- In Fig. 2, the authors say there are four layers. However, I do not see any layers on this figure. Besides, I have no idea why the authors address IoT devices on this figure since it seems that they have nothing to do with the east-west communication.
- The authors introduce \rho as reputation of a controller. The reputation is computed based on the controller's audit compliance, policy enforcement accuracy and network uptime. Note that they are controllers and their policies are defined by the administrator. So it is not easy for me to see why controllers have different reputations. So I cannot see the value of algorithm 2.
- It is not clear for me to see the contribution of algorithm 3. Except the signature part, it looks like a general federated learning architecture.
- The authors use some machine learning metrics to evaluate their work. The experimental results show that the this work can effectively detect malicious nodes, which are sensors. However, I cannot see the relation between this experiment with the east-west communication. For me, this is just a machine-learning research which integrates CNN and SVM.
Fine.
Author Response
The reviewer responses file is attached below.

Reviewer 2 Report
Comments and Suggestions for Authors
Major comments
The article presents a malicious node detection model BFL-SDWanTrust to solves the issues of centralized malicious node detection schemes and third-party-based multi-controller SD-WANs. The distributed model training was ensured at edge nodes without central data aggregation. The distributed model training enhances not only the accuracy of the malicious node detection process but also preserves the privacy of network entities. The authors also propose a Blockchain-based network that validates all network communication and malicious node detection transactions without the involvement of any third party.
There is no need to make any improvements other than correcting minor shortcomings pointed out below. The authors have briefly outlined the purpose of the study, given a detailed description of the materials and method, conducted an analytical review of many scientific articles, and formulated answers to the questions posed.
Detailed comments
Provide a paragraph describing each section of the manuscript at the end of Introduction.
Write ‘where’ instead ‘Where’ after equations.
Add section Discussion.
Describe where the obtained results can be applied in the end of Conclusions.
The manuscript has many abbreviations. I recommend that the authors provide an Abbreviation section.
Provide appropriate text in the missing paragraphs Author Contributions, Funding, and Conflicts of Interest.
References should be prepared exactly in accordance with the journal template. References in recent manuscripts published in the journal could also help you with it.
Author Response

(The authors gave the same response as above.)

Reviewer 3 Report
Comments and Suggestions for Authors
Firstly, Algorithm 4 are excessively complex and burdened with repetitive cryptographic operations, such as hashing and digital signatures at nearly every architectural layer (sensing, blockchain, fog, and cloud). While these operations are individually justifiable, their frequent and layered repetition introduces computational redundancy. The manuscript does not provide any analysis of their cumulative impact on performance metrics such as latency, throughput, or energy efficiency, nor does it demonstrate that these cryptographic operations significantly contribute to the observed improvements in detection accuracy or trust validation. This lack of justification undermines the practical efficiency of the proposed system.
In Equation (6), the authors define the validator selection probability as a weighted function of trust reputation and historical transaction success rate , modulated by weights ω₁ and ω₂, respectively. The paper refers to Reference [35] (Zhao et al., 2024) as the origin of this formulation. However, a closer examination of Zhao et al.'s work, "Blockchain-Based Security Deployment and Resource Allocation in SDN-Enabled MEC System", reveals that their context, model assumptions, and network architecture differ significantly from the SD-WAN scenario presented in this manuscript
Author Response

(The authors gave the same response as above.)

Reviewer 4 Report
Comments and Suggestions for Authors
The authors of this quite interesting paper propose a malicious node detection model, BFL-SDWanTrust, that solves the issues of centralized malicious node detection schemes and third-party-based multi-controller SD-WANs. The issues of privacy leakage, performance bottlenecks and single points of failure were also solved while utilizing the capabilities of federated learning and blockchain techniques.
The subject of the paper is interesting, well presented and analyzed. The related work is adequate and the same comment applies to the simulation results presented by the authors. Some minor syntax/grammar errors (see e.g. line 53 "There model..." ) can be corrected during the preparation of the camera ready version.
Author Response

(The authors gave the same response as above.)

Round 2
Reviewer 1 Report
Comments and Suggestions for Authors
All of my comments are addressed. However, I think there are still many serious problems. My comments are as followed.
- The authors use InSDN dataset to replace TON_IOT and it sounds reasonable. However, I think InSDN does not include east-west communication data. Actually, in InSDN paper, where the citation is as follows,
M. S. Elsayed, N. -A. Le-Khac and A. D. Jurcut, "InSDN: A Novel SDN Intrusion Dataset," in IEEE Access, vol. 8, pp. 165263-165284, 2020, doi: 10.1109/ACCESS.2020.3022633, "
Mahmoud Said Elsayed et al. claim that due to the hardware constraints, the low scale topology with only one SDN controller was considered and implemented. This implies there is no east-west communication in this dataset. Moreover, the authors claim there are five types of attacks in InSDN, including ARP spoofing. I do not see ARP spoofing attack in InSDN paper. Besides, according to InSDN paper, the dataset is generated with ONOS, not POX. - I do not see any convincing east-west attacks description in this paper. The authors claim that there are many possible attacks between controllers like rogue controller impersonation. As described in my previous comments, TLS supports mutual authentication and therefore, I cannot see how impersonation attacks are launched.
- The authors build a controller reputation system. The authors claim that the reputation of a controller is based on network policies, responsiveness, uptime and so on. However, according to Fig. 2, these controllers are managed by a logical centralized controller. So why these controllers have different policies? Even they have different policies, they are set by the global controller.
- If the authors switch the dataset from TON_IoT to InSDN, why TON_IoT is shown in Fig. 15?
Author Response

(The authors gave the same response as above.)

Reviewer 3 Report
Comments and Suggestions for Authors
All issues have been addressed and all questions have been answered. The paper is now much clearer. I would suggest a few minor edits to the English, but nothing critical. I wish the authors success in their future work.
Author Response

(The authors gave the same response as above.)
